# Evolution Strategies at Scale:
# LLM Fine-Tuning Beyond Reinforcement Learning

**Xin Qiu** [* † 1]  **Yulu Gan** [* ‡ 2]  **Conor F. Hayes** [* 1]  **Qiyao Liang** [‡ 2]  **Yinggan Xu** [‡ 3]  **Roberto Dailey** [1]
**Elliot Meyerson** [1]  **Babak Hodjat** [1]  **Risto Miikkulainen** [1 4]

## Abstract

Fine-tuning large language models (LLMs) for downstream tasks is an essential stage of modern AI deployment. Reinforcement learning (RL) has emerged as the dominant fine-tuning paradigm, underpinning many state-of-the-art LLMs. In contrast, evolution strategies (ES) has largely been overlooked due to the widespread belief that it does not scale to modern model sizes. This paper overturns this assumption by demonstrating the first successful application of ES to full-parameter fine-tuning of LLMs at the billion-parameter scale, without dimensionality reduction. ES can indeed search over extremely high-dimensional parameter spaces and outperform established RL implementations across multiple axes, including improved tolerance to long-horizon and delayed rewards, robustness across diverse base LLMs, reduced susceptibility to reward hacking, and improved training stability. These findings suggest that ES is not merely a viable alternative to RL, but a fundamentally different and powerful backpropagation-free post-training paradigm that opens a new direction for LLM fine-tuning beyond current RL-based approaches.

## 1. Introduction

As the capabilities of large language models (LLMs) have rapidly improved, these systems have been increasingly deployed across scientific and engineering workflows (Touvron et al., 2023; Achiam et al., 2024; AI@Meta, 2024; Jiang et al., 2024; Liu et al., 2024a; Anthropic, 2025; Google,

2025; Singhal et al., 2023; Wu et al., 2023; Rozière et al., 2024; Romera-Paredes et al., 2024). This widespread deployment has made fine-tuning a standard step for adapting pre-trained models to downstream tasks and aligning behavior with user preferences (Ouyang et al., 2022; Rafailov et al., 2023; Latif & Zhai, 2024; Guo et al., 2025a). In practice, reinforcement learning (RL) has become the predominant choice for this fine-tuning stage (Ouyang et al., 2022; Bai et al., 2022; Shao et al., 2024; Guo et al., 2025a;b; Srivastava & Aggarwal, 2025). However, several challenges have emerged: First, RL methods incur low sample efficiency and high variance of the gradient estimator when handling long-horizon rewards, which is a common case for LLM fine-tuning with outcome-only rewards (Salimans et al., 2017; Sutton & Barto, 2018; Vemula et al., 2019). Proper credit assignment at token level for RL fine-tuning methods is difficult and possibly unhelpful (Zhang et al., 2025; Song et al., 2025; Guo et al., 2025b; Uesato et al., 2022; Jia et al., 2025; Guo et al., 2025b). Second, RL techniques are sensitive to the choice of base LLMs, resulting in inconsistent fine-tuning performance across different models (Gandhi et al., 2025). Third, RL techniques tend to incentivize hacking the reward function, leading to undesirable behaviors (Gao et al., 2023; Denison et al., 2024; Fu et al., 2025). Fourth, RL fine-tuning is often unstable across multiple runs even with the same hyperparameter settings, significantly increasing fine-tuning cost (Choshen et al., 2020; Zhong et al., 2025).

Evolution Strategies (ES), a class of population-based zeroth-order optimization algorithms, is a possible alternative. ES has several advantages over RL in traditional control and gaming problems: it parallelizes naturally, tolerates long-horizon rewards, promotes broad exploration, avoids expensive backpropagation, and remains robust across hyperparameter settings (Salimans et al., 2017; Chrabaszcz et al., 2018; Conti et al., 2018). However, ES remains relatively underexplored in LLM fine-tuning settings. Standard ES directly optimizes in the full parameter space, which in prior applications typically contained no more than a few million parameters (Salimans et al., 2017; Zhang et al., 2017; Lehman et al., 2018; Lorenc & Neruda, 2025). It was

---

[*]Equal contribution . [†]Project Lead. [‡]Work done during an internship at Cognizant AI Lab. [1]Cognizant AI Lab, San Francisco, CA, USA [2]Massachusetts Institute of Technology, Cambridge, MA, USA [3]University of California, Los Angeles, Los Angeles, CA, USA [4]The University of Texas at Austin, Austin, TX, USA. Correspondence to: Xin Qiu <qiuxin.nju@gmail.com>.

*Proceedings of the 43rd International Conference on Machine Learning*, Seoul, South Korea. PMLR 306, 2026. Copyright 2026 by the author(s).

---

Source code: https://github.com/VsonicV/es-fine-tuning-paper

assumed that for very large models, exploration in parameter space is significantly more difficult and sample-inefficient than exploration in action space (Vemula et al., 2019). Modern LLMs typically contain billions of parameters, which makes direct ES optimization appear infeasible. Existing workarounds include restricting ES to the final layer of the base model (Toledano-López et al., 2022), applying ES to low-dimensional adapters (Jin et al., 2024), and performing evolutionary search in action space, analogous to standard RL (Huang et al., 2025). Directly searching in the full parameter space of LLMs (without dimensionality reduction) has remained a challenge.

This paper is aimed at meeting this challenge. For the first time, ES is scaled to multi-billion-parameter search spaces through direct optimization of the full parameter space of LLMs during fine-tuning. The approach is based on a memory-efficient implementation of an algorithmically simplified ES variant, with support for parallelization across GPUs. Performance is compared with state-of-the-art (SOTA) RL methods in fine-tuning various LLMs in several reasoning benchmark tasks, and behavioral differences from RL are analyzed in terms of fine-tuning for conciseness. Furthermore, ES fine-tuning is successfully applied to solve two puzzle problems that are challenging for base LLMs.

ES was found able to search directly over billions of parameters without dimensionality reduction while achieving strong fine-tuning performance relative to RL in multiple aspects: (1) *ES only needs response-level rewards*, making it a perfect fit for fine-tuning on reasoning tasks that have only sparse long-horizon outcome rewards. In particular, ES obtained significantly better fine-tuned models than RL in the Countdown task with such rewards. (2) *ES is able to find good solutions in large space with small populations*, e.g. just 30 in the multi-billion-parameter space in this paper. As a comparison, previous ES implementations (Salimans et al., 2017; Zhang et al., 2017; Lehman et al., 2018; Lorenc & Neruda, 2025) utilized a population size of 10,000 or more with much smaller models (i.e. millions of parameters or less). The current extremely small population size thus makes the approach feasible even without extensive compute. (3) *ES is more robust* than RL across different LLMs. While RL fine-tuning failed on some LLMs, ES provided good fine-tuning for all of them. ES benefits from its exploration in parameter space, making it less sensitive to initial states of the LLMs. (4) *ES consistently maintains reasonable behaviors* during fine-tuning, in contrast to RL that tends to hack the reward function if no other penalty is added. The main reason is that ES optimizes a solution distribution (Lehman et al., 2018), which is more difficult to hack, while RL optimizes a single solution. (5) *ES's behavior is more consistent* than RL's across different runs. This property can significantly reduce expected cost of fine-tuning. (6) *Fine-tuning with ES only requires inference*, and

therefore no gradient calculations are needed. A significant amount of GPU memory can therefore be saved.

Thus, this study establishes a critical first milestone in demonstrating that ES can serve as a viable and powerful post-training paradigm for LLMs. The results reveal a surprising and counterintuitive finding that ES remains effective when scaled to models with billions of parameters, directly challenging the long-held assumption that such methods are inherently unscalable. These findings not only motivate further scaling to even larger LLMs, but fundamentally expand the design space of post-training algorithms. By operating directly in parameter space without reliance on gradients or intermediate supervision, ES enables new forms of outcome-only optimization, robust exploration over high-dimensional parameter landscapes, and naturally distributed large-scale fine-tuning. Taken together, this paper positions ES as a foundational alternative to gradient-based RL and opens a new direction for scalable, stable, and general LLM post-training.

## 2. Related Work

The background on Evolution Strategies and evolutionary optimization of LLMs is first reviewed, followed by SOTA RL fine tuning and parameter-space exploration.

**Traditional ES:** Evolution Strategies (ES, Rechenberg, 1973; Schwefel, 1977) are a class of evolutionary algorithms (EAs) for solving numerical optimization problems. The main idea is to sample a population of solutions through perturbations, then recombine the perturbed solutions based on their fitness values to form the population for the next generation. This process repeats until a termination condition is triggered, e.g., the maximum number of generations is reached. Among the different variants of ES, CMA-ES (Hansen & Ostermeier, 2001), which utilizes a multivariate Gaussian distribution with full covariance matrix to sample the population, and natural ES (Wierstra et al., 2008; 2014), which uses natural gradient to guide the search, are two popular methods for traditional optimization problems. Although ES has long been used to evolve parameters of neural networks (NNs), (Igel, 2003), Salimans et al. (2017) were the first to scale the approach up to deep learning networks. Comparable performance to RL methods in control and gaming environments was observed, and several unique advantages of ES highlighted. This seminal work paved the way for several follow-up studies. Zhang et al. (2017) used ES to optimize a convolutional NN with around three million parameters. They found that with a large enough population size, ES can approximate the performance of traditional stochastic gradient descent (SGD). Lehman et al. (2018) further optimized a NN comprising nearly 167,000 parameters with both ES and a finite-difference (FD) gradient estimator. Because ES optimizes the average reward for

the entire population, whereas FD optimizes the reward for a single solution, it obtained models that were more robust to parameter perturbations. Lorenc & Neruda (2025) applied ES to optimize decision transformers in RL environments, and observed promising results for model sizes up to around 2.5 million parameters. In a related study, another traditional EA, namely genetic algorithm (GA) with mutations only, was extended to a high-dimensional space (Such et al., 2017). Encouraging results were observed in different types of models with up to around four million parameters (Such et al., 2017; Risi & Stanley, 2019). However, although these studies were promising, the scale of these implementations was still significantly less than the size of current LLMs.

**Evolution+LLMs:** Synergies between Evolutionary Algorithms (EAs) and LLMs have received increasing attention in recent years (Wang et al., 2025; Wu et al., 2025). Popular research directions include EAs for prompt optimization (Sun et al., 2022b;a; Zhao et al., 2023; Guo et al., 2024), utilizing LLMs as evolutionary operators (Meyerson et al., 2024; Lehman et al., 2024; Romera-Paredes et al., 2024; Novikov et al., 2025), and merging LLMs through evolution (Du et al., 2024; Akiba et al., 2025). Applying EAs to optimize billions of parameters in LLMs is generally perceived to be intractable, but a few studies have been successful at a smaller scale. For example, Toledano-López et al. (2022) fine-tuned the last layer (with 325 parameters) of an mT5-based transformer via CMA-ES. Jin et al. (2024) optimized the low-rank adapter parameters (with dimensionality up to 1600) using CMA-ES and the Fireworks algorithm. Sanchez Carmona et al. (2024) applied a GA to fine-tune around 9.5 million parameters of a transformer encoder, though poorer performance than the traditional Adam optimizer was observed. Huang et al. (2025) proposed a hybrid algorithm that performs exploration in action space instead of parameter space, and it was only used in the final epoch of supervised fine-tuning (SFT). The work in this paper significantly extends this prior research by successfully scaling ES to search in the billions of parameters of LLMs, leading to surprisingly good fine-tuning performance.

**RL for fine-tuning:** Fine-tuning using RL is a critical step during the training of many landmark LLMs (Ouyang et al., 2022; Bai et al., 2022; Shao et al., 2024; Guo et al., 2025a;b). Proximal Policy Optimization (PPO; Schulman et al., 2017) and Group Relative Policy Optimization (GRPO; Shao et al., 2024) are the two predominant methods. PPO introduces a clipped surrogate objective to limit the update scale in each step with respect to the old policy, and it usually works with a value model in an actor-critic manner. GRPO simplifies the pipeline of PPO by replacing the value model with group advantage, which is calculated based on direct evaluations of multiple responses. As discussed in Section 1, in the context of LLM fine-tuning, these methods struggle with several fundamental limitations, including the dilemma in handling

long-horizon reward (Vemula et al., 2019; Salimans et al., 2017; Zhang et al., 2025; Song et al., 2025; Uesato et al., 2022; Jia et al., 2025; Guo et al., 2025b), sensitivity to base LLMs (Gandhi et al., 2025), tendency to hack reward (Gao et al., 2023; Denison et al., 2024; Fu et al., 2025), and instability across runs (Choshen et al., 2020; Zhong et al., 2025). ES inherently avoids these limitations, leading to better fine-tuning performance.

**Parameter-space exploration:** Existing RL fine-tuning methods are overwhelmingly based on action-space exploration. Parameter space exploration has received much less attention, though some such studies do exist (Rückstieß et al., 2008; Sehnke et al., 2010; Rückstieß et al., 2010; Plappert et al., 2018). Although promising performance was observed in problems with sparse rewards, the scale of the tested models was far smaller than that of LLMs. Vemula et al. (2019) performed a theoretical analysis of different exploration strategies, and found that the complexity of the parameter space exploration increased quadratically with the number of parameters, whereas the complexity of action space exploration depended on action dimensionality quadratically and horizon length of the reward quartically. Based on the classical SPSA method (Spall, 1992), Malladi et al. (2023) proposed a zeroth-order optimizer MeZO that directly worked in parameter space for fine-tuning LLMs. MeZO significantly reduced memory requirements, but its fine-tuning performance was no better than other baselines. In contrast, the ES implementation in this paper performs exploration in multi-billion-parameter search spaces, and exhibits strong performance across different benchmarks.

## 3. Method

This section introduces the basic ES algorithm and a detailed description of its implementation for LLM fine-tuning.

### 3.1. Basic ES algorithm

---

**Algorithm 1** Basic ES Algorithm

---

**Require:** Pretrained LLM with initial parameters $\boldsymbol{\theta}_0$, reward function $R(\cdot)$, total iterations $T$, population size $N$, noise scale $\sigma$, learning rate $\alpha$.

1: **for** $t = 1$ to $T$ **do** $\qquad\qquad$ ▷ outer ES iterations
2: $\quad$ **for** $n = 1$ to $N$ **do**
3: $\quad\quad$ Sample noise $\boldsymbol{\varepsilon}_n \sim \mathcal{N}(0, \boldsymbol{I})$
4: $\quad\quad$ Compute reward for perturbed parameters:
$\quad\quad\quad R_n \leftarrow R(\boldsymbol{\theta}_{t-1} + \sigma \cdot \boldsymbol{\varepsilon}_n)$
5: $\quad$ **end for**
6: $\quad$ Normalize $R_n$
7: $\quad$ Update model parameters as
$\quad\quad \boldsymbol{\theta}_t \leftarrow \boldsymbol{\theta}_{t-1} + \alpha \cdot \frac{1}{N} \sum_{n=1}^{N} R_n \boldsymbol{\varepsilon}_n$
8: **end for**

---

---

**Algorithm 2** ES Implementation for LLM Fine-Tuning

---

**Require:** Pretrained LLM with initial parameters $\boldsymbol{\theta}_0$, reward function $R(\cdot)$, total iterations $T$, population size $N$, noise scale $\sigma$, learning rate $\alpha$, number of parallel process $P$.

1: Create $P$ processes, each instantiates a model with the same initial parameters $\boldsymbol{\theta}_0$, with one process as the main process
2: **for** $t = 1$ to $T$ **do**                                                        $\triangleright$ ES iterations
3:     Sample N random seeds $s_1, s_2, \ldots, s_N$
4:     Assign random seeds to $P$ processes
5:     **for** $n = 1$ to $N$ **do**
6:         For the process handling $s_n$, reset its random number generator using random seed $s_n$
7:         **for each LLM layer do**                               $\triangleright$ perturbation within current process
8:             Sample noise $\boldsymbol{\varepsilon}_{n,l} \sim \mathcal{N}(0, \boldsymbol{I})$, which has the same shape as the $l$th layer's parameters
9:             Perturb the $l$th layer's parameters in-place: $\boldsymbol{\theta}_{t-1,l} \leftarrow \boldsymbol{\theta}_{t-1,l} + \sigma \cdot \boldsymbol{\varepsilon}_{n,l}$
10:         **end for**
11:         Compute reward for perturbed parameters $R_n \leftarrow R(\boldsymbol{\theta}_{t-1})$                $\triangleright$ within current process
12:         For the process handling $s_n$, reset its random number generator using random seed $s_n$
13:         **for each LLM layer do**                            $\triangleright$ restoration within current process
14:             Sample noise $\boldsymbol{\varepsilon}_{n,l} \sim \mathcal{N}(0, \boldsymbol{I})$, which has the same shape as the $l$th layer's parameters
15:             Restore the $l$th layer's parameters in-place: $\boldsymbol{\theta}_{t-1,l} \leftarrow \boldsymbol{\theta}_{t-1,l} - \sigma \cdot \boldsymbol{\varepsilon}_{n,l}$
16:         **end for**
17:     **end for**
18:     Normalize the reward scores by calculating the $z$-score for each $R_n$: $Z_n \leftarrow \frac{R_n - R_{\mathrm{mean}}}{R_{\mathrm{std}}}$,
        where $R_{\mathrm{mean}}$ and $R_{\mathrm{std}}$ are the mean and standard deviation of $R_1, R_2, \ldots, R_N$.
19:     **for** $n = 1$ to $N$ **do**                                             $\triangleright$ in main process only
20:         Reset current random number generator using random seed $s_n$
21:         **for each LLM layer do**
22:             Sample noise $\boldsymbol{\varepsilon}_{n,l} \sim \mathcal{N}(0, \boldsymbol{I})$, which has the same shape as the $l$th layer's parameters
23:             Update $l$th layer's parameters in-place as $\boldsymbol{\theta}_{t,l} \leftarrow \boldsymbol{\theta}_{t-1,l} + \alpha \cdot \frac{1}{N} Z_n \boldsymbol{\varepsilon}_{n,l}$
24:         **end for**
25:     **end for**
26:     Update the model parameters of all processes to $\boldsymbol{\theta}_t$
27: **end for**

---

The ES implementation used in this paper is based on a simplified variant of Natural Evolution Strategies (NES) (Wierstra et al., 2008; 2014) and follows the design of OpenAI ES (Salimans et al., 2017), which employs fixed-covariance perturbation noise.

Given a pretrained LLM with initial parameters $\boldsymbol{\theta}_0$ and a target reward function $R(\cdot)$, the task is to fine-tune the parameters so that the reward function is optimized (Algorithm 1). In each iteration, $N$ perturbed models are sampled by adding random Gaussian noise $\boldsymbol{\varepsilon}_n$ to their parameters. The noise is i.i.d. in each dimension of the parameter space, and it is scaled by the hyperparameter $\sigma$. The perturbed models are evaluated to obtain their reward scores $R_n$. The final update of the model parameters aggregates the sampled perturbations by weighting them using their normalized reward scores. The standard update equation $\boldsymbol{\theta}_t \leftarrow \boldsymbol{\theta}_{t-1} + \alpha \cdot \frac{1}{\sigma} \frac{1}{N} \sum_{n=1}^{N} R_n \boldsymbol{\varepsilon}_n$ is simplified to $\boldsymbol{\theta}_t \leftarrow \boldsymbol{\theta}_{t-1} + \alpha \cdot \frac{1}{N} \sum_{n=1}^{N} R_n \boldsymbol{\varepsilon}_n$ by digesting the term $\frac{1}{\sigma}$ into the learning rate $\alpha$.

To improve scalability, a number of modifications to this basic algorithm were made as detailed in the next section.

### 3.2. Implementation details

The actual implementation of ES for this paper expands on the basic ES algorithm in seven ways (see Algorithm 2 for the detailed pseudocode):

(1) *Noise retrieval with random seeds:* Similar to Salimans et al. (2017); Such et al. (2017), only the random seeds are stored to reduce GPU memory usage. The perturbation noise used during sampling can be retrieved exactly by resetting the random number generator with specific random seeds. (2) *Parallel evaluations:* In each iteration, the perturbed models can be evaluated fully in parallel by assigning a separate random seed to each process. (3) *Layer-level in-place perturbation and restoration:* To reduce the peak GPU memory usage, the model parameters are perturbed in-place layer by layer, with corresponding random seeds archived. After evaluation of the perturbed model, the model parameters are restored by subtracting the same

| Base Model | Original | RL | | | | | ES (ours) |
|---|---|---|---|---|---|---|---|
| | | PPO-z | GRPO-z (8) | GRPO-z (30) | GRPO-v | Dr.GRPO-v | |
| Qwen-2.5-0.5B-Instruct | 0.1 | 0.3 | 0.3 | 0.5 | 13.0 | 13.5 | **14.4** |
| Qwen-2.5-1.5B-Instruct | 0.7 | 14.2 | 13.9 | 14.8 | 27.8 | 31.0 | **37.3** |
| Qwen-2.5-3B-Instruct | 10.0 | 20.1 | 30.9 | 32.5 | 37.8 | 43.8 | **60.5** |
| Qwen-2.5-7B-Instruct | 31.2 | 55.1 | 54.2 | 52.8 | 57.0 | 57.5 | **66.8** |
| Llama-3.2-1B-Instruct | 0.4 | 11.2 | 14.5 | 13.0 | 14.9 | 13.9 | **16.8** |
| Llama-3.2-3B-Instruct | 3.2 | 35.3 | 39.4 | 38.8 | 42.5 | 47.8 | **51.6** |
| Llama-3.1-8B-Instruct | 8.1 | 42.8 | 49.9 | 51.3 | 46.9 | 50.2 | **61.2** |

*Table 1.* Accuracy (%) on the Countdown task across model families, sizes, and fine-tuning algorithms. Different model families are shaded for clarity; *Original* refers to directly evaluating the base model without any fine-tuning, and GRPO-z (8) and GRPO-z (30) indicate group sizes of 8 and 30. The suffix "-z" and "-v" represents different implementation variants (see Appendix A.1 for more details). The same hyperparameters were used for all ES runs; a separate grid search for the best hyperparameters was run for each RL experiment.

noise perturbations using the archived random seeds. For each evaluation process, apart from the model parameters, the only additional memory needed is to store a tensor the size of a layer temporarily. (4) *Reward normalization:* The rewards of the perturbed models are normalized using $z$-score within each iteration, so that the normalized rewards for each iteration have a mean of 0 and standard deviation of 1. This normalization makes the reward scale consistent across iterations and tasks. (5) *Greedy decoding:* The perturbed models use greedy decoding to generate the responses for reward evaluations. As a result, the perturbed models are evaluated deterministically, so that all performance differences come from the exploration in parameter space instead of action space. (6) *Decomposition of the parameter update:* At the end of each iteration, the aggregated update of model parameters is performed in-place in a decomposed manner, gradually adding up layer by layer and seed by seed, significantly reducing the peak GPU memory needed. (7) *Learning rate digestion:* The standard update equation $\theta_t \leftarrow \theta_{t-1} + \alpha \cdot \frac{1}{\sigma} \frac{1}{N} \sum_{n=1}^{N} R_n \varepsilon_n$ is simplified to $\theta_t \leftarrow \theta_{t-1} + \alpha \cdot \frac{1}{N} \sum_{n=1}^{N} R_n \varepsilon_n$ by digesting the term $\frac{1}{\sigma}$ into the learning rate $\alpha$, simplifying the computation and parametric setup.

To highlight the strength of ES, we intentionally remove common algorithmic enhancements explored in OpenAI ES (Salimans et al., 2017). Enhancements like rank transformation of rewards (Wierstra et al., 2014), mirrored sampling (Sehnke et al., 2010), weight decay, and virtual batch normalization (Salimans et al., 2016) are not used in this work. Additionally, we do not utilize more advanced optimizers like Adam (Kingma & Ba, 2015). This design choice isolates the core ES algorithm and demonstrates that strong performance can be achieved without auxiliary enhancements. In future work, each individual enhancement can be explored to further improve performance.

## 4. Empirical Studies

This section first compares the fine-tuning performance of ES and RL baselines on a standard reasoning benchmark. After that, behavioral differences between ES and RL are investigated in fine-tuning for conciseness, followed by comparisons to more SOTA RL baselines on several math reasoning tasks. Finally, ES is applied to solve two challenging puzzle problems.

### 4.1. Performance in the Countdown task

Fine-tuning performance was measured in the Countdown task (Gandhi et al., 2024; Pan et al., 2025), a symbolic reasoning benchmark (see Appendix A.3 for details), showing that ES is accurate and efficient across different kinds and sizes of LLMs, even when the RL approaches are not.

**Experimental setup.** A single fixed set of hyperparameters ($N = 30$, $\sigma = 0.001$, $\alpha = 5 \times 10^{-4}$) was used for all ES Countdown experiments. Notably, the population size 30 is significantly lower than those in previous works (Salimans et al., 2017; Zhang et al., 2017), in which $N \geq 10,000$. For RL baselines (see Appendix A.1 for details), a separate hyperparameter sweep was done for each experiment. RL methods turned out sensitive to hyperparameters, in particular the KL-divergence penalty coefficient $\beta$ and learning rate $\alpha$, and did not make much progress if they were not set precisely. To mitigate this issue, for each model, a small grid of $\beta$ and $\alpha$ values were tested and the best-performing configuration selected (see Table 4 in the Appendix A.1). This approach makes the comparison conservative with respect to ES, but it also highlights its robustness. **ES improves upon RL baselines across all tested models.** Previously, Gandhi et al. (2025) found that RL does not generalize well across models on the Countdown task. Table 1 confirms this result, and also demonstrates that ES does not have this problem. With each model in the Qwen2.5 family (0.5B–7B) and the Llama3 family (1B–8B), ES substantially improved over

| Model | $\beta$ | $\alpha$ | $\sigma$ | Reward $\uparrow$ | KL $\downarrow$ |
|---|---|---|---|---|---|
| Qwen-2.5-7B+GRPO | 0.0 | $5 \times 10^{-6}$ | ✗ | $0.867 \pm 0.054^*$ | $0.861 \pm 0.614^*$ |
| Qwen-2.5-7B+GRPO | 0.01 | $5 \times 10^{-6}$ | ✗ | $0.871 \pm 0.060^*$ | $1.354 \pm 0.873^*$ |
| Qwen-2.5-7B+GRPO | 0.0167 | $5 \times 10^{-6}$ | ✗ | $0.911 \pm 0.038$ | $1.591 \pm 0.811$ |
| Qwen-2.5-7B+GRPO | 0.0464 | $5 \times 10^{-6}$ | ✗ | $0.881 \pm 0.062$ | $1.384 \pm 1.187$ |
| Qwen-2.5-7B+ES | ✗ | 0.0005 | 0.001 | $0.889 \pm \mathbf{0.004}$ | $0.274 \pm \mathbf{0.096}$ |
| Qwen-2.5-7B+ES | ✗ | 0.00075 | 0.0015 | $0.919 \pm \mathbf{0.008}$ | $0.813 \pm \mathbf{0.212}$ |

*Table 2.* Behavior or GRPO and ES in terms of mean conciseness reward and mean KL divergence. The label $^*$ indicates cases where reward hacking was observed. Only models that did not hack the reward were included in the results.

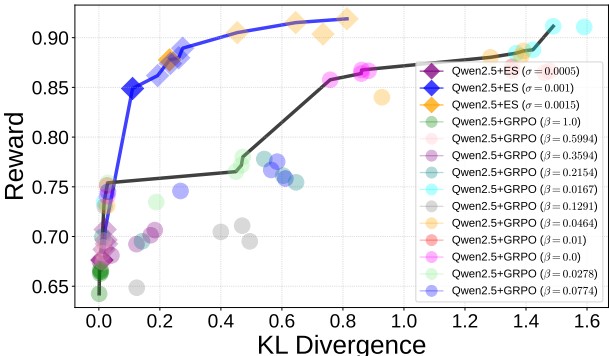

*Figure 1.* Mean conciseness reward and mean KL divergence from the base model for each fine-tuning checkpoint across different learning parameters. The Pareto front of ES (blue line) is higher and to the left of the GRPO Pareto front (black line) models, indicating that it found better tradeoffs. ES discovers these solutions without any KL divergence penalty, suggesting that it represents a distinctly different fine-tuning mechanism from the GRPO.

PPO, GRPO and Dr.GRPO (Liu et al., 2025b), including their implementation variants, often by a large margin (see Figure 6 in Appendix A.6 for a model-wise visual comparison). Additional experiments on performance variances across multiple runs are provided in Appendix A.4, showing a clear performance separability between ES and RL. These results demonstrate that ES scales effectively across different model types and sizes, and does so significantly better than RL.

### 4.2. Behavioral differences between ES and RL in fine-tuning for conciseness

In order to characterize the different approaches that ES and RL take, they were used to fine-tune Qwen-2.5-7B Instruct, towards more concise responses in question-answering (see Appendix A.1 for more details). That is, fine-tuning was rewarded based on how concise the answers were, but not directly rewarded for its question-answering performance. In this setup, it was possible to analyze not only whether fine-tuning was effective, but also how it was achieved, including what its side effects were.

**ES discovers a dominant Pareto front.** Similarly to Rafailov et al. (2023), a Pareto frontier analysis was used to compare ES and GRPO, with mean reward and mean KL divergence as the metrics (Figure 1). The experimental setup is described in Appendix A.1. The ES Pareto front is represented by a blue line on top and the GRPO Pareto front by the black line below. That is, ES produced better tradeoffs than GRPO, i.e. models with higher reward and lower KL divergence. The GRPO results were achieved only after augmenting the conciseness reward with a KL divergence penalty (weighted by a parameter $\beta$). Without it, fine-tuning resulted in excessive divergence and incorrect answers. Remarkably, ES achieved superior tradeoffs without any KL divergence penalty, suggesting that ES fine-tuning is based on discovering distinctly different kinds of solutions than GRPO. Appendix A.5 presents additional experiments with varying $\alpha$ and $\beta$ values, yielding similar conclusions.

**ES is more robust against reward hacking.** GRPO with $\beta = \{0.0, 0.01\}$ sometimes hacked the reward. Reward hacking was detected quantitatively using the KL divergence estimator defined in Schulman (2020). A model was automatically marked as a "possible anomaly" if the KL divergence exploded by at least one order of magnitude. For these "possible anomaly" models, response details were checked to confirm the reward hacking behavior. Models which hacked the reward typically produced responses that were short but contain nonsensical symbols rather than words. By increasing the KL-penalty via higher $\beta$ values, reward hacking could be prevented. The optimal $\beta$ is likely to be problem specific and to require extensive search to find. In contrast, ES does not receive any feedback about the divergence of the fine-tuned model, and only seeks to optimize conciseness. Regardless, it did not exhibit any reward hacking, despite achieving mean reward comparable to GRPO with $\beta = \{0.0, 0.01\}$. This result again suggests that ES finds a different way of optimizing the reward function.

**ES fine-tuning is reliable across runs.** Fine-tuning LLMs is computationally expensive, so it is critical that it leads to consistent results across runs. Table 2 presents the mean and standard deviation of the conciseness reward and KL divergence across four independent runs after $1,000$ iterations.

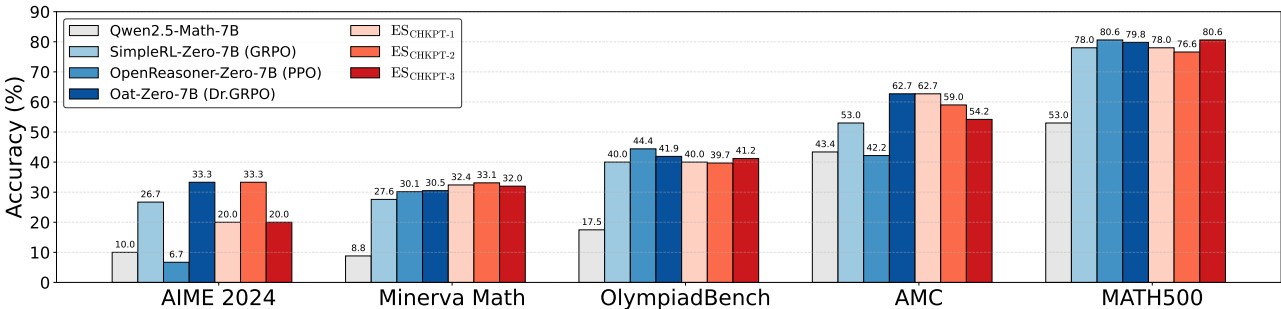

*Figure 2.* Performance of ES compared to strong, well-established RL baselines across math reasoning benchmarks. Across all benchmarks, ES achieved competitive performance compared to OpenReasoner-Zero-7B (PPO), Simple-RL-Zero (GRPO), Oat-Zero-7B (Dr.GRPO). Given the vanilla nature of the current ES implementation, these results constitute a promising starting point for ES fine tuning in math.

A mean reward cut-off of $> 0.85$ was used to down-select hyperparameter combinations, ensuring that only the best ES and GRPO configurations were included in the analysis. From Table 2, ES achieved consistent conciseness rewards, indicated by a low standard deviation over four runs with different random seeds. GRPO has $15.5\times$ higher standard deviation, suggesting that its results were much less consistent. The results on KL divergence show similar patterns. Thus, ES fine-tuning is more reliable than GRPO.

### 4.3. ES applied to Math reasoning tasks

RL has been shown to enhance the reasoning capabilities of LLMs through post-training with verifiable rule-based rewards. To understand the impact of ES on LLM reasoning, ES fine-tuning was evaluated on a set of standard math benchmarks from the literature. The main result is that ES is competitive with SOTA RL in this setting.

**Training setup.** The Qwen2.5-Math-7B (Yang et al., 2024) base model was fine-tuned with ES using the MATH dataset (Hendrycks et al., 2021). Problems labeled with difficulty ranging from 3-5 were included, and the Qwen Math template was used for training with ES (see Appendix A.7; Yang et al., 2024). Both RL and ES sampled a maximum of $3,000$ tokens per response; ES hyperparameters were set to $\sigma = 0.001$, $\alpha = \frac{\sigma}{2}$, and $N = 30$.

**RL baselines.** The fine-tuned ES models were compared with strong, well-established baselines from the literature. These RL implementations achieve the most SOTA performance in the tested benchmarks, utilizing production-ready RL libraries like VERL (Sheng et al., 2024) and OAT (Liu et al., 2024b). They include SimpleRL-Zero (GRPO) (Zeng et al., 2025), OatZero (Dr.GRPO) (Liu et al., 2025b), and OpenReasoner (PPO) (Hu et al., 2025). The publicly released Qwen2.5-7B checkpoints trained with the original training recipes were used for evaluation. Note that SimpleRL-Zero and OatZero were trained using the MATH dataset (Hendrycks et al., 2021), whereas OpenReasoner

was trained using a custom dataset compiled by the authors. Consequently, performance differences should be interpreted in light of both algorithmic and dataset-related differences.

**Evaluation benchmarks.** Several standard math reasoning benchmarks from the literature are used for evaluation: OlympiadBench (He et al., 2024), MATH500 (Hendrycks et al., 2021), Minerva (Lewkowycz et al., 2022), AIME2024 (Li et al., 2024), and AMC (Li et al., 2024). The pass@1 accuracy metric was used in the evaluations.

**Key results.** Figure 2 shows the performance of the base model, three checkpoints of RL baselines, and three checkpoints of ES (see Appendix A.7 for more details). ES significantly improved the base models across each benchmark, showing the other optimization methods aside from RL can be used to elicit improvement in LLM reasoning capabilities. In addition, ES exhibits competitive performances compared with the SOTA RL baselines in all the benchmarks. It is notable that these RL baselines are the best-performing implementations selected from the literature, with extensive algorithmic refinement and hyperparameter search particularly for math reasoning tasks. In contrast, the current ES implementation is a vanilla variant with a simple hyperparameter setup; thus, the results constitute a promising starting point for the ES approach in math fine tuning.

### 4.4. Solving challenging puzzle problems

To further evaluate the generality of ES in tackling different types of tasks, two challenging puzzle problems were used as additional testbeds. The first is ARC-AGI (Chollet et al., 2024), a benchmark designed to evaluate fluid intelligence (Chollet, 2019). The second is Sudoku, a logic-based number-placement puzzle. Whereas the base LLM models fail severely in both problems, ES fine-tuning significantly improves their performance (Table 3). Experimental details are provided in Appendix A.11 for ARC-AGI and Appendix A.10 for Sudoku.

| Task | Base Model | Original | ES Fine-tuned |
|------|-----------|----------|---------------|
| ARC-AGI | Qwen-2.5-14B | 0.2 | **29.5** |
| Sudoku | Qwen-2.5-3B | 2.5 | **69.5** |

*Table 3.* Accuracy (%) on solving puzzle problems. *Original* refers to directly evaluating the base model without any fine-tuning. ES fine-tuning improves performance significantly, demonstrating that it can be applied to a range of problems.

## 5. Discussion

**Algorithmic advantage of ES vs. RL.** Exploration in parameter space plays a key role in the surprisingly good fine-tuning performance of ES. As discussed by Rückstieß et al. (2010) and Plappert et al. (2018), sampling noise in parameter space ensures that the entire action trajectory, i.e., the sequence of tokens, only depends on one single sampling, leading to significantly lower variance in rollouts, i.e., in response generation. As a result, gradient estimation is more reliable and convergence is more stable. In contrast, action space exploration in RL injects noise at every step, i.e., at each token position, resulting in high variance in the sequence generation. The behavior of RL therefore is much less reliable than ES, as was seen in Table 2. Moreover, step-wise exploration in action space promotes reward hacking by increasing the chance of sampling a single hacking action. One example is the nonsensical symbol sampled during RL that can hack the conciseness reward.

Another key difference between ES and RL is that ES intrinsically optimizes a solution distribution (Lehman et al., 2018), while RL optimizes a single solution. This property makes it more difficult for ES to hack the reward since a single hacked solution usually does not have a high-quality solution distribution around it. This property also results in solutions that are more robust to noisy perturbations in parameter space (Lehman et al., 2018), making them more robust to adversarial attacks and less likely to be compromised in other follow-up fine-tuning tasks (Chen et al., 2025).

The ES algorithm presented in Algorithm 2 is very simple and easy to implement, without need for sophisticated hyperparameter search. In the current experiments, ES with a single fixed set of hyperparameters was able to achieve robust performances across different tasks and base models. Ablation studies in Appendix A.2 further demonstrate the insensitivity of ES to hyperparameters. In contrast, RL algorithms are considerably more complex and require substantial expertise to implement robustly across tasks and systems, usually with extensive hyperparameter tuning. In particular, there has been significant debate in the literature regarding best practices for implementing GRPO. Effective GRPO implementations typically rely on a number of non-obvious design choices and implementation details, such as removing length normalization (Liu et al., 2025b) and using

more aggressive clipping (Yu et al., 2025). Many of these practices have only emerged through extensive empirical investigation. Moreover, the application of the KL penalty in GRPO remains an open design choice, with alternatives such as applying it to the loss or directly to the reward leading to markedly different performance outcomes (Shah et al., 2025)

**Engineering benefits of ES vs. RL.** Modern RL frameworks grow increasingly complex as they are applied to LLMs with ever-increasing parameter counts. Deploying these systems in practice often requires substantial engineering effort and computational resources. In contrast, ES is simple to implement and can help democratize post-training by significantly lowering engineering and systems overhead. This section outlines two key advantages of ES and suggests how it can be scaled to fine-tune the largest LLMs.

(1) *Parallelization.* To minimize memory overhead and maximize sample throughput, RL systems rely on asynchronous architectures in which actors are distributed across GPUs and update a shared learner model. While effective, scaling these systems across large numbers of GPUs and computational nodes introduces significant engineering complexity. In contrast, as shown by Salimans et al. (2017), ES can be trivially parallelized: as the number of available GPUs increases, the population size can be scaled accordingly.

Modern frontier AI research labs operate clusters with thousands of GPUs[1], making efficient large-scale parallelization possible. While it is challenging to do with RL, ES requires only the exchange of random seeds (noise) and scalar rewards between machines. Such simple communication enables parallel ES to be used either to reduce wall-clock training time or to scale to much larger populations.

(2) *Gradient computation.* Asynchronous RL makes it possible to compute actor-related gradients in parallel. However, in order to manage memory usage, gradient checkpointing and multiple learner updates per synchronization step are needed. While these techniques enable larger effective batch sizes, they also require gradients to be communicated across GPUs, and sometimes nodes, introducing significant memory overhead and engineering complexity. This complexity scales with both the number of GPUs and the size of the model, and is further exacerbated when model parameters must be sharded across devices.

In contrast, ES does not require gradient computation. By eliminating gradient calculation and communication entirely, ES avoids much of the associated engineering and memory overhead. As a result, each member of the ES population can use large batch sizes freely without cross-device gradient synchronization, which can yield substantial practical and performance benefits. Moreover, the computational

---

[1]LLama3 training used $16,000$ H100 GPUs (AI@Meta, 2024).

cost in terms of FLOPs is significantly lower for ES compared to RL methods, which involve backpropagation and additional reference models (See Appendx A.8 for detailed analysis).

Importantly, ES is an inference-only fine-tuning mechanism, where the model weights are never differentiated, only evaluated. This property opens the door to specialized inference kernels optimized for repeated forward passes, large batches, and parameter perturbations. These mechanisms are difficult to leverage in gradient-based training regimes, but are possible in ES fine tuning in the future.

## 6. Future Work

One counterintuitive result is that the ES implementation only needs a population of 30 to effectively optimize billions of parameters. In contrast, previous work (Salimans et al., 2017; Zhang et al., 2017; Lehman et al., 2018; Lorenc & Neruda, 2025) used populations of 10,000 or more for models with millions or fewer parameters. An interesting future direction is to analyze how such small populations are possible. Perhaps this is related to the observed low intrinsic dimensionality of LLMs (Aghajanyan et al., 2021). Another promising direction is to use ES to perform unsupervised fine-tuning based on internal behaviors of LLMs, such as confidence calculated based on semantic entropy and semantic density (Qiu & Miikkulainen, 2024; Farquhar et al., 2024). Such fine-tuning cannot be done with RL, since action space exploration does not change the internal representations of LLMs (that is, each action sampling is generated via output distribution without changing the internal parameters). In a broader sense, since ES does not need process rewards during exploration, it may be a necessary ingredient for superintelligence (Mucci & Stryker, 2023), which would be difficult to achieve by supervised learning using process guidance from human data. Massive parallelization of ES will speed up exploration by distributing the computations across GPU machines or even data centers.

An important question is: what are the underlying computational mechanisms that make ES and RL behave so differently? While this question requires significant further work, a possible hypothesis emerges from the experiments in this paper. Many fine-tuning objectives, like conciseness and the Countdown task, are long-horizon outcome-only objectives. The reward signal is jagged, making it difficult to navigate with gradient-based post-training methods. RL and ES both provide workarounds via effective noise injection to "smooth out" the jagged reward landscape. In the case of RL, noise is introduced from Monte-Carlo sampling of each token during a rollout, averaged over many rollouts, which effectively smooths the sampling process but does not necessarily guarantee that the reward landscape is smooth

in parameter space. RL's gradient estimation therefore has a high-variance, and its signal-to-noise ratio becomes worse with longer sequences and sharper policies (i.e. those with lower entropy), and therefore prone to undesirable outcomes such as reward hacking.

In contrast, ES injects noise directly into the parameter space via explicit Gaussian convolution, which effectively smooths out the jagged reward landscape. As a result, it provides a more stable way of exploring the landscape, leading to more consistent, efficient, and robust optimization (as observed in the experiments and in Appendix A.9). Moreover, the larger the models and the sharper the policies, the more jagged the reward landscapes; therefore, ES is likely to have an advantage in fine-tuning them. Direct evidence for this hypothesis still needs to be obtained, but it provides a plausible mechanistic explanation, and a direction for future work.

In principle, ES and RL are not mutually exclusive. They can be complementary to each other and used in a combined way. One possibility is to divide the learning process into exploration and exploitation phases, and then alternate applying ES and RL. For global exploration, perturbation-based parameter space search is more explorative and less likely to get stuck in an initial local optimum. For fine-grained local search, following the gradients is a more accurate and effective approach. Investigating synergies between ES and RL could result in better fine-tuning methods, as well as an improved understanding of LLM training in general.

## 7. Conclusion

This paper introduces a fundamentally new paradigm for fine-tuning LLMs by scaling ES to models with billions of parameters without dimensionality reduction. Contrary to long-standing assumption that such scaling is infeasible, the paper demonstrates that ES can efficiently fine-tune the full parameter space of modern LLMs and, in doing so, consistently surpasses standard RL-based fine-tuning methods. On the Countdown task, with sparse long-horizon rewards challenging for gradient-based RL, ES achieves substantially stronger performance. It also exhibits markedly reduced sensitivity to hyperparameter choices and delivers stable, repeatable improvements across multiple base LLMs. In fine-tuning for conciseness, ES is less prone to reward hacking and shows reliable behavior across independent runs. The generality of ES fine tuning is further validated by strong performance on state-of-the-art math reasoning benchmarks and two challenging puzzle problems. Together, these results establish ES as a scalable, robust, and general fine-tuning method, and demonstrate that backpropagation-free optimization can serve as a powerful alternative to RL for fine-tuning LLMs.

## Impact Statement

Beyond the standard potential consequences of advancing the field of machine learning, there are two key areas of broader impact, stemming from (1) *increased ease of use* and (2) *reduced reward-hacking*.

**Ease of use:** ES qualitatively reduces the barrier of entry to fine-tuning LLMs. Unlike RL, which requires an expert mathematical understanding of nuanced gradient-based training dynamics to design an effective reward function, ES simply requires the experimenter to assign *a score* to a model after it has attempted a task. This simplification democratizes LLM fine-tuning, opening the door to the development of customized AI applications by non-experts.

**Reward-hacking:** As shown in Section 4.2 and prior work (Lehman et al., 2018), ES is inherently less susceptible to reward-hacking than RL and other gradient-based methods. Thus, LLMs fine-tuned with ES are less likely to lose ethical guardrails present in the base model. Similarly, it may be easier to fine-tune for ethical behavior (i.e. alignment) with ES, since the model is less likely to overfit to specific training examples.

Combining the above two areas of impact, ES fine-tuning can reduce the risk of unintended ethical misbehavior of LLMs fine-tuned by non-experts.

## Acknowledgments

We would like to thank Sid Stuart for providing technical support for hardware management and always being responsive. We would like to thank Kajetan Schweighofer for his generous help during the peer-review rebuttal phase and insightful discussions. We would like to thank Jamieson Warner for providing valuable feedback.

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

# A. Appendix

## A.1. Experimental Setup

**Experimental setup for the Countdown experiments.** Representative models from the Qwen2.5 family (0.5B–7B) and the Llama3 family (1B–8B) were fine-tuned for this task. For the PPO-z experiments, a grid search was first performed around common hyperparameter settings and the best-performing values used (Table 4). TinyZero (https://github.com/Jiayi-Pan/TinyZero) is used for PPO-z implementations. For the GRPO-z experiments, a grid search was performed around the settings of (Pan et al., 2025) and the best-performing values used. GRPO-z experiments were run with two different group sizes: $N = 8$, following the common practice in GRPO training for the Countdown task, and $N = 30$, aligning with the population size in ES. GRPO-Zero (https://github.com/policy-gradient/GRPO-Zero) is used for GRPO-z implementations. VERL (Sheng et al., 2024) is used for both GRPO-v and Dr.GRPO-v implementations, with the standard default configurations for math reasoning benchmarks.

For the VERL implementations, we set the global batch size of 1024, a learning rate of $1 \times 10^{-6}$, and a rollout group size of $N = 8$. We compared two configurations: GRPO-v and Dr.GRPO-v. The GRPO-v baseline incorporated a standard KL divergence penalty with a coefficient of $\beta = 0.001$. In contrast, the Dr.GRPO-v configuration removed the KL penalty (use_kl_loss=False) and disabled advantage normalization (norm_adv_by_std=False). Instead, Dr.GRPO-v employed a sequence-mean token-sum normalization strategy for loss aggregation with a scaling factor of 1024.

For all the ES and RL baselines, the total number of sample evaluations was the same. The ES population size was $N = 30$, noise scale $\sigma = 0.001$, and learning rate $\alpha = 5 \times 10^{-4}$ across all experiments. To evaluate accuracy, a set of 200 samples were used during training, and a different set of 2000 samples during testing. For ES, results were reported on the test set after training for 500 iterations. For RL, the training was stopped after the same total number of sample evaluations as in the ES runs. An example of the prompt and the response is provided in Appendix A.3.

| Method | Model | (1e−3, 1e−6) | (1e−3, 1e−5) | (5e−3, 1e−6) | (5e−3, 1e−5) |
|---|---|---|---|---|---|
| PPO-z | Qwen-0.5B-Instruct | ✓ | | | |
| | Qwen-1.5B-Instruct | ✓ | | | |
| | Qwen-3B-Instruct | ✓ | | | |
| | Qwen-7B-Instruct | | ✓ | | |
| | Llama-1B-Instruct | | ✓ | | |
| | Llama-3B-Instruct | | | | ✓ |
| | Llama-8B-Instruct | | | ✓ | |
| GRPO-z | Qwen-0.5B-Instruct | | ✓ | | |
| | Qwen-1.5B-Instruct | | | ✓ | |
| | Qwen-3B-Instruct | | ✓ | | |
| | Qwen-7B-Instruct | ✓ | | | |
| | Llama-1B-Instruct | | | | ✓ |
| | Llama-3B-Instruct | | ✓ | | |
| | Llama-8B-Instruct | ✓ | | | |

*Table 4.* Hyperparameter Sweep across Models under PPO-z and GRPO-z. Each pair $(\cdot, \cdot)$ denotes (KL-divergence penalty coefficient $\beta$, learning rate $\alpha$); the label '✓' indicates the best hyperparameter setting for each model-method combination.

**Experimental setup for the conciseness experiments.** In each experiment, Qwen-2.5-7B-Instruct (Yang et al., 2025) was fine-tuned using both ES and GRPO and evaluated using a held-out evaluation set. Each run was repeated four times, using a different random seed each time. For each GRPO experiment, the group size $N = 30$, and learning rate $\alpha = 5 \times 10^{-6}$. Ten log-spaced values from 0.01 to 1.0 were evaluated for the the KL-divergence penalty coefficient $\beta$, as well as $\beta = 0.0$. Appendix A.5 presents additional experiments with varying $\alpha$ and $\beta$ values. For ES, the population size $N = 30$, ensuring that GRPO and ES generated the same number of responses per prompt, resulting in the same training exposure. Models were fine-tuned with $\sigma = \{0.0005, 0.001, 0.0015\}$, with a learning rate $\alpha = \frac{\sigma}{2}$. Both GRPO and ES experiments were run for $1,000$ iterations, and a checkpoint saved every 200 iterations. Table 5 shows the dataset of prompts and verifiable solutions used during fine-tuning; note that it consists of only two examples. Similarly, Table 6 lists the prompts and verifiable solutions used in evaluating each fine-tuned model. For all the experimental results, the displayed reward values

| Prompt | Verifiable Solution |
|---|---|
| Solve: $3 + 5 =$ | 8 |
| If all birds can fly and penguins are birds, can penguins fly | No |

*Table 5.* Prompts and verifiable solutions used in fine-tuning the models for conciseness. Two examples is enough to achieve this goal.

| Prompt | Verifiable Solution |
|---|---|
| What is the capital of France? | Paris |
| Calculate: $12 \times 7 =$ | 84 |
| Is the statement "All cats are mammals" true or false? | True |
| What comes next in the sequence: $2, 4, 6, 8, ?$ | 10 |
| Translate "Hello" to Spanish: | Hola |
| What is $15\%$ of 200? | 30 |
| Name one primary color: | Red |
| How many days are in a week? | 7 |

*Table 6.* Prompts and verifiable solutions used to evaluate the fine-tuned models. More examples are necessary than during fine-tuning to make the evaluation reliable.

are normalized to be within $[0, 1]$, with 0 corresponding to $-2000$ in the original reward function and 1 corresponding to the best possible original reward 0.

**Conciseness task.** For conciseness fine-tuning, a dataset of prompts $\mathcal{D} = \{x_1, .., x_K\}$, with a set of verifiable solutions $\{s_1, ..., s_K\}$, i.e. shortest possible correct answers, was used. For example, for the prompt "Name one primary color", possible shortest verifiable solution used is "Red". Following this approach, for each prompt $x \in \mathcal{D}$, the model was encouraged to generate a concise response $y$. To fine-tune the model to generate concise responses, a reward computed using the absolute length difference between the generated response $y$ and the corresponding verified solution $s_k$ was given to the model for each prompt $x_k$. The reward function $R$ for conciseness was defined as $R = -|\text{len}(y) - \text{len}(s_k)|$, where $\text{len}(\cdot)$ denotes the string length.

**Behavior metrics for the conciseness experiments.** Behavior of the fine-tuned models was measured in two ways: the mean conciseness reward and the mean KL divergence from the base model (after Rafailov et al., 2023). KL divergence is useful as a proxy for the preservation of the base model's behavior. It correlates strongly with the question-answering performance of the model, but also conveys more information, i.e. the extent of the fine-tuning changes. A low KL divergence thus suggests that the fine-tuned model has not forgotten capabilities learned during pre-training. Further, as KL divergence increases, these capabilities are likely to break. Therefore, fine-tuning behavior can be characterized using the tradeoffs between reward and KL divergence. To compute the metrics, each fine-tuned model was evaluated on a set of held-out test prompts, with 20 responses sampled per prompt. The reward was computed using the model-generated response and the verifiable solution provided in the test dataset. The KL divergence between a fine-tuned model $\theta_{\text{FT}}$ and a base model $\theta_{\text{BASE}}$ for a given prompt $x$ and corresponding response $y$ was approximated following Schulman (2020) as $\text{KL}[\theta_{\text{FT}} \parallel \theta_{\text{BASE}}] = \frac{\theta_{\text{BASE}}(y_{i,t}|x, y_{i,<t})}{\theta_{\text{FT}}(y_{i,t}|x, y_{i,<t})} - \log \frac{\theta_{\text{BASE}}(y_{i,t}|x, y_{i,<t})}{\theta_{\text{FT}}(y_{i,t}|x, y_{i,<t})} - 1$.

**Partially correlated noise vs. i.i.d. noise.** In the current implementation, for each perturbed model, the random number generator is reinitialized for each layer (using the random seed corresponding to this model), leading to partially correlated noise perturbations across layers. Preliminary experiments show that this implementation does not lead to significantly different performances compared to true i.i.d. noise.

**A.2. ES Hyperparameter Ablations**

To further investigate how ES performance varies with population size $N$, noise scale $\sigma$, and learning rate $\alpha$, a number of ablation experiments were conducted in the GSM8K reasoning benchmark using Qwen2.5-1.5B and Qwen2.5-3B. Each experiment ran for 300 training steps, with ES sampling a maximum of 512 tokens per prompt. All runs used a fixed training dataset of 512 prompts and were evaluated on the GSM8K test set using pass@1 with greedy decoding.

**Effect of population size ($N$) on performance.** First, the effect of varying the population size ($N$) was studied by setting $N \in \{16, 30, 64, 128\}$. Table 7 and Table 8 present the pass@1 scores of the final models. Across Qwen2.5-1.5B and Qwen2.5-3B, larger population sizes generally yielded better pass@1 performance, however, the differences were modest. Since increasing $N$ incurs additional computational cost, $N = 30$ emerged as a reasonable tradeoff, balancing performance and computational efficiency.

| $N$ | 16 | 30 | 64 | 128 |
|---|---|---|---|---|
| pass@1 | 0.697 | 0.714 | 0.725 | 0.727 |

*Table 7.* Pass@1 scores for population size ($N$) ablation for Qwen2.5-1.5B on GSM8K. Larger $N$ generally leads to better performance, but the differences are moderate.

| $N$ | 16 | 30 | 64 | 128 |
|---|---|---|---|---|
| pass@1 | 0.807 | 0.823 | 0.804 | 0.831 |

*Table 8.* Pass@1 scores for population size ($N$) ablation for Qwen2.5-3B on GSM8K. $N = 128$ gives the best performance, but $N = 30$ also provides good performance at lower cost.

**Effect of noise scale ($\sigma$) and learning rate ($\alpha$) on performance.** Next, $\sigma$ and $\alpha$ were varied to study their effect on performance. Table 9 and 10 present the final model pass@1 score on GSM8K for each $\sigma$ and $\alpha$ configuration across Qwen2.5-1.5B and Qwen2.5-3B. The choice of $\sigma$ and $\alpha$ had a moderate effect on the final performance of both models. Qwen2.5-1.5B with $\sigma = 0.0005$ and $\alpha = 0.00025$ and Qwen2.5-3B with $\sigma = 0.001$ and $\alpha = 0.001$ produced the worst final performance, while $\sigma = 0.002$ and $\alpha = 0.0005$ achieved the best performance across both models. Both Tables 9 and 10 highlight that ES is robust across different hyperparameter settings and show that ES maintains similar performance across all tested configurations.

| $\sigma$ | $\alpha$ | pass@1 |
|---|---|---|
| 0.001 | 0.0005 | 0.714 |
| 0.002 | 0.001 | 0.688 |
| 0.0005 | 0.00025 | 0.654 |
| 0.002 | 0.0005 | 0.726 |
| 0.0005 | 0.0005 | 0.675 |
| 0.001 | 0.001 | 0.660 |
| 0.001 | 0.00025 | 0.704 |

*Table 9.* Pass@1 scores for $\sigma$ and $\alpha$ ablation for Qwen2.5-1.5B on GSM8K. ES is robust across different hyperparameter settings.

| $\sigma$ | $\alpha$ | pass@1 |
|---|---|---|
| 0.001 | 0.0005 | 0.817 |
| 0.002 | 0.001 | 0.808 |
| 0.0005 | 0.00025 | 0.792 |
| 0.002 | 0.0005 | 0.826 |
| 0.0005 | 0.0005 | 0.791 |
| 0.001 | 0.001 | 0.782 |
| 0.001 | 0.00025 | 0.823 |

*Table 10.* Pass@1 scores for $\sigma$ and $\alpha$ ablation for Qwen2.5-3B on GSM8K. ES maintains similar performance across different hyperparameter settings.

### A.3. Examples of Countdown Task Outputs

**Countdown task.** The Countdown task (Gandhi et al., 2024; Pan et al., 2025) requires constructing an arithmetic expression from a given set of numbers using basic operations ($+, -, \times, \div$) to match a target value. For instance, the target 950 can be

obtained from $\{100, 50, 6, 3\}$ with $100 \times (6 + 3) + 50 = 950$). This constitutes a compact test of constrained symbolic reasoning, i.e. an important use case for fine-tuning.

---

**Iteration 400**

**[1] Input:**

You are a helpful assistant. You first think about the reasoning process in your mind and then provide the user with the answer. Using the numbers [49 41 73], create an equation that equals 17. You can use basic arithmetic operations (+, −, *, /) and each number can only be used once. Show your work in `<think>`... `</think>` tags. And return the final answer in `<answer>`... `</answer>` tags, for example `<answer> (1 + 2) / 3 </answer>`.

**[1] Response:**

`<think>` First, let's consider all possible combinations of the numbers 49, 41, and 73 using basic arithmetic operations. We need to use each number exactly once and combine them to get 17.
1. Start with simple addition and subtraction: - 49 + 41 - 73 = 80 - 73 = 7 (too high) - 49 - 41 + 73 = 8 + 73 = 81 (too high) - 73 - 49 - 41 = 24 - 41 = -17 (too low)
2. Try multiplication and division: - (49 * 41) / 73 = 2029 / 73 ≈ 27.83 (not close to 17) - (73 * 41) / 49 = 3013 / 49 ≈ 61.47 (too high) - (49 * 73) / 41 = 3637 / 41 ≈ 88.80 (too high) - (41 * 73) / 49 = 3013 / 49 ≈ 61.47 (same as above)
3. Check different orders: - 73 - (49 + 41) = 73 - 90 = -17 (too low) - 49 - (73 - 41) = 49 - 32 = 17 (this looks promising)
So, the equation 49 - (73 - 41) equals 17. Let's verify it: - Calculate inside the parentheses first: 73 - 41 = 32 - Then subtract from 49: 49 - 32 = 17
This satisfies our requirement. `</think>` `<answer>` 49 - (73 - 41) `</answer>`
Reward: 1.1000 (Format: 1.0000, Answer: 1.0000)

*Figure 3.* An example of a countdown task interaction.

---

Figure A.3 shows an example of the Qwen2.5-3B model's output on the Countdown task after 400 ES iterations. In order to improve accuracy, the model learned to adjust its formulas by comparing the generated answer with the correct one, and to verify its own answers for correctness.

## A.4. Additional Results for the Countdown Experiments

To further verify the variance across runs for the Countdown task, three independent runs with different random seeds were conducted for ES, GRPO-v and Dr.GRPO-v. The experimental setup was the same as in Section A.1. As shown in Table 11, ES had the lowest variances across multiple runs. In an additional Student's $t$-test, the performance differences between ES and other GRPO variants are statistically significant with $p < 0.05$.

| Mean ± Std | ES | GRPO-v | DR.GRPO-v |
|---|---|---|---|
| Qwen2.5-1.5B-Instruct | $37.73 \pm 1.14$ | $26.77 \pm 1.98$ | $32.97 \pm 2.05$ |
| Qwen2.5-3B-Instruct | $58.67 \pm 1.47$ | $35.07 \pm 2.33$ | $44.43 \pm 1.89$ |

*Table 11.* Performance (accuracy) statistics over three independent runs on the Countdown Task. ES exhibits the lowest variance across runs, and the performance differences between ES and GRPO variants are statistically significant.

## A.5. Extended Conciseness Details and Experiments

In this section, the conciseness experiments are extended to investigate the impact of different learning rates on GRPO training.

**GRPO with different learning rates.** Further GRPO experiments were run over four seeds with $\beta = \{0, 0.01, 0.1, 1.0\}$, varying the learning rate $\alpha = \{2 \times 10^{-6}, 3 \times 10^{-6}, 4 \times 10^{-6}, 5 \times 10^{-6}\}$. A total of 20 responses were sampled per evaluation prompt. Figure 4a shows the mean reward and KL divergence of each fine-tuned model. As the learning rate increases, both mean reward and mean KL divergence increase. The best models with respect to reward are trained using $5 \times 10^{-6}$ and $\beta = \{0.0, 0.01\}$, obtaining rewards greater than 0.85. Figure 4b further displays the GRPO Pareto front (black

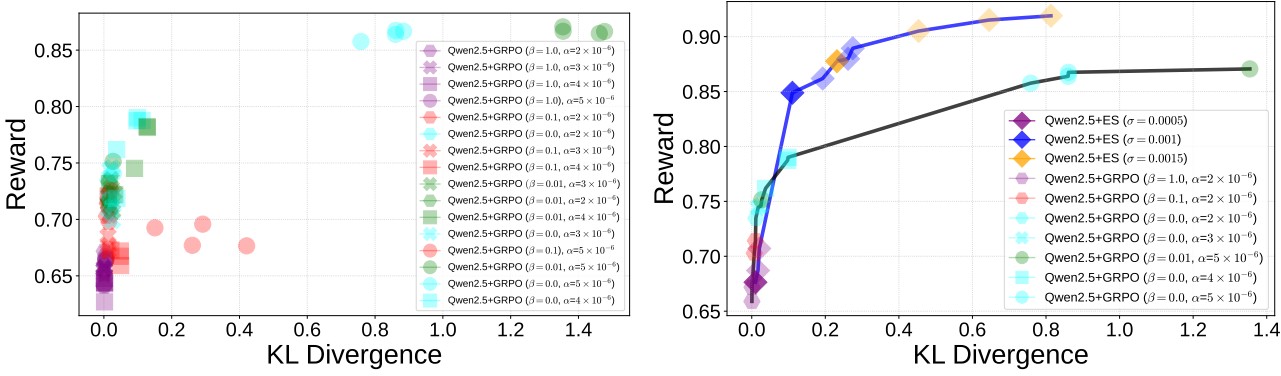

*(a)* GRPO models results over various learning rates.   *(b)* ES and GRPO Pareto fronts.

*Figure 4.* GRPO behavior with different learning rates. (a) GRPO models trained using different learning rates and $\beta$ values. Both conciseness reward and KL divergence increase with higher learning rates. (b) The ES Pareto front (blue line, top) plotted with the GRPO Pareto front (black line, bottom) over different model learning parameters. ES dominates GRPO across the whole range.

line, bottom) across these learning rates, comparing it with the ES Pareto front (blue line, top). The majority of Pareto optimal models across these learning rates obtain a mean reward of less than $0.8$ and a KL divergence of less than $0.4$. The ES Pareto front dominates that of GRPO over different learning rates and $\beta$ values.

Next, the reward distribution for each $\alpha$ and $\beta$ value for GRPO was compared with that of ES, starting with learning rates $2 \times 10^{-6}$ and $3 \times 10^{-6}$. Figures 5a and Figure 5b show that all GRPO models stay close to the Qwen2.5-7B-Instruct base model reward distribution, despite the variation in $\beta$. In contrast, ES shifts the reward distribution to the right with a density peak around $1.0$, i.e. towards higher rewards. The learning rate was then further increased to $4 \times 10^{-6}$ (Figure 5c). As a result, for $\beta = 0.0$ and $\beta = 0.01$, GRPO shifts the reward distribution to the right towards higher rewards. However, they are still lower than those of ES. As the learning rate is increased further to $5 \times 10^{-6}$ (Figure 5d), GRPO is sufficiently able to optimize the reward: with $\beta = 0.0$ and $\beta = 0.01$, it peaks around $1.0$. Thus, high learning rate combined with low $\beta$ is important for GRPO to optimize the reward. However, as was discussed before, such a setting often breaks the performance of the model.

### A.6. Training Curves and Accuracy Improvement of ES and RL on the Countdown Task

As shown in Figure 7, ES consistently outperformed RL across all tested models throughout training. In addition, as shown in Figure 6, we compute the relative improvements of PPO, GRPO, DR.GRPO and ES over their respective base models across different model families. ES delivers the consistently largest improvements in all cases.

### A.7. Extended Math Reasoning Details and Discussion

**RL baselines.**   We compare ES against three strong R1-Zero-style (Guo et al., 2025a) reasoning baselines at the 7B parameter scale: SimpleRL-Zero (Zeng et al., 2025), OpenReasoner-Zero (Hu et al., 2025), and Oat-Zero (Liu et al., 2025b). All baselines are instantiated using Qwen2.5-series models (Yang et al., 2025), which are competitive open-weight language models known to exhibit strong reasoning performance at this scale. The respective baseline implementations are fully open source and built on production-ready RL libraries, including VERL (Sheng et al., 2024) and OAT (Liu et al., 2024b), which provided highly optimized, stable, and tested PPO, GRPO, and Dr.GRPO implementations with efficient rollout management, and standardized reward handling. SimpleRL-Zero isolates the core R1-Zero optimization mechanism with minimal additional engineering, OpenReasoner-Zero reflects a widely adopted community implementation with practical design choices, and Oat-Zero further alters GRPO with algoritmic enhancements shown to boost performance. Together, these baselines represent strong, reproducible, and non-trivial comparators for evaluating ES.

**Reward function.**   Our ES training utilizes a basic rule-based reward function that checks answer correctness, without any format rewards. The reward function is designed to extract the produced answer contained within \boxed{} and compare it with the ground truth answer. Similarly to RL, we implement a binary reward scheme where a reward of 1 is given for exact matches with the reference answer, and 0 for all other cases. To ensure a fair comparison with models from the literature we use the same answer extractor, also called a grader, as OatZero (Liu et al., 2025b).

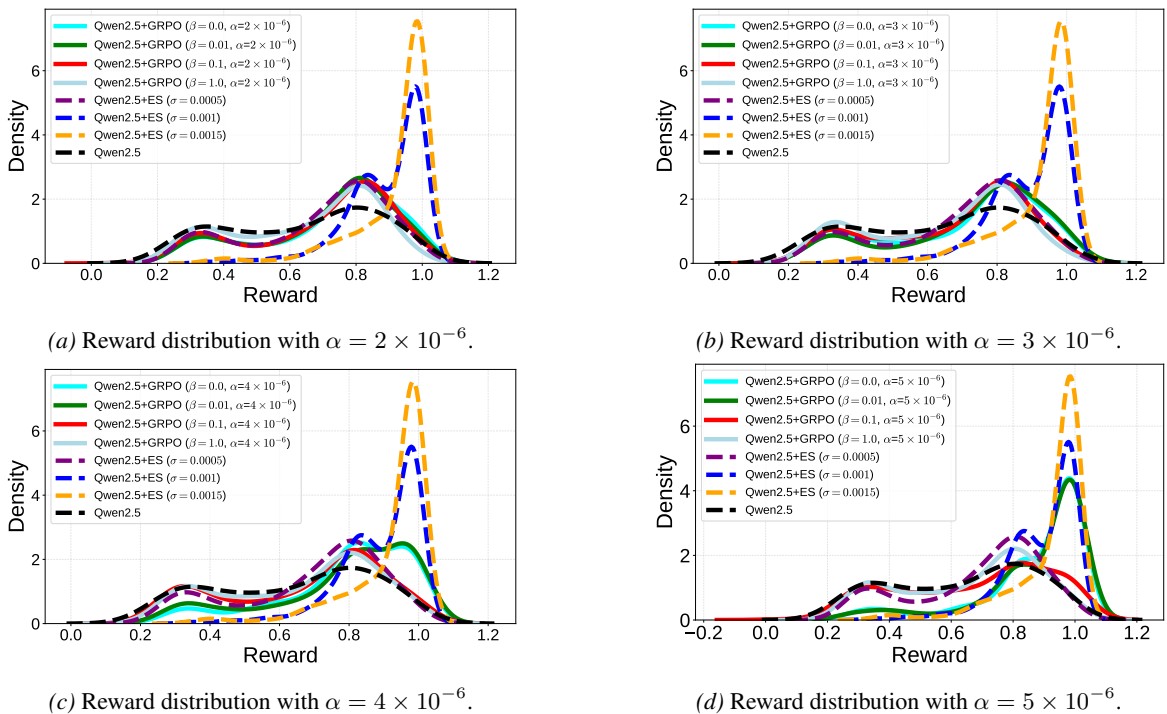

*(a)* Reward distribution with $\alpha = 2 \times 10^{-6}$.

*(b)* Reward distribution with $\alpha = 3 \times 10^{-6}$.

*(c)* Reward distribution with $\alpha = 4 \times 10^{-6}$.

*(d)* Reward distribution with $\alpha = 5 \times 10^{-6}$.

*Figure 5.* Reward distributions in fine-tuning for conciseness with different learning rates $\alpha = \{2 \times 10^{-6}, 3 \times 10^{-6}, 4 \times 10^{-6}, 5 \times 10^{-6}\}$ and $\beta = \{0.0, 0.01, 0.1, 1.0\}$ compared to ES on the Qwen2.5-7B-Instruct base model. Whereas GRPO distribution is similar to the base model, ES shifts it to the right, i.e. higher rewards. Higher rewards can only be achieved with GRPO with high learning rates and low $\beta$, which setting often breaks to model's performance.

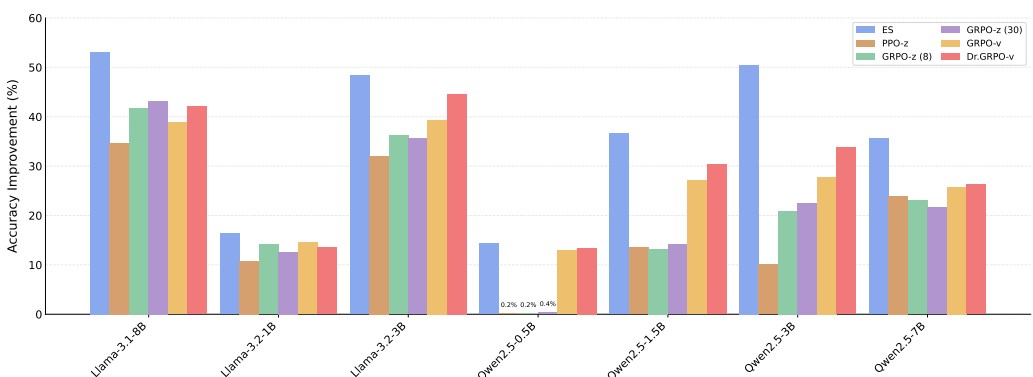

*Figure 6.* Accuracy Improvement over Base Models with ES vs RL across Model Families. ES results in consistently largest improvements in all cases.

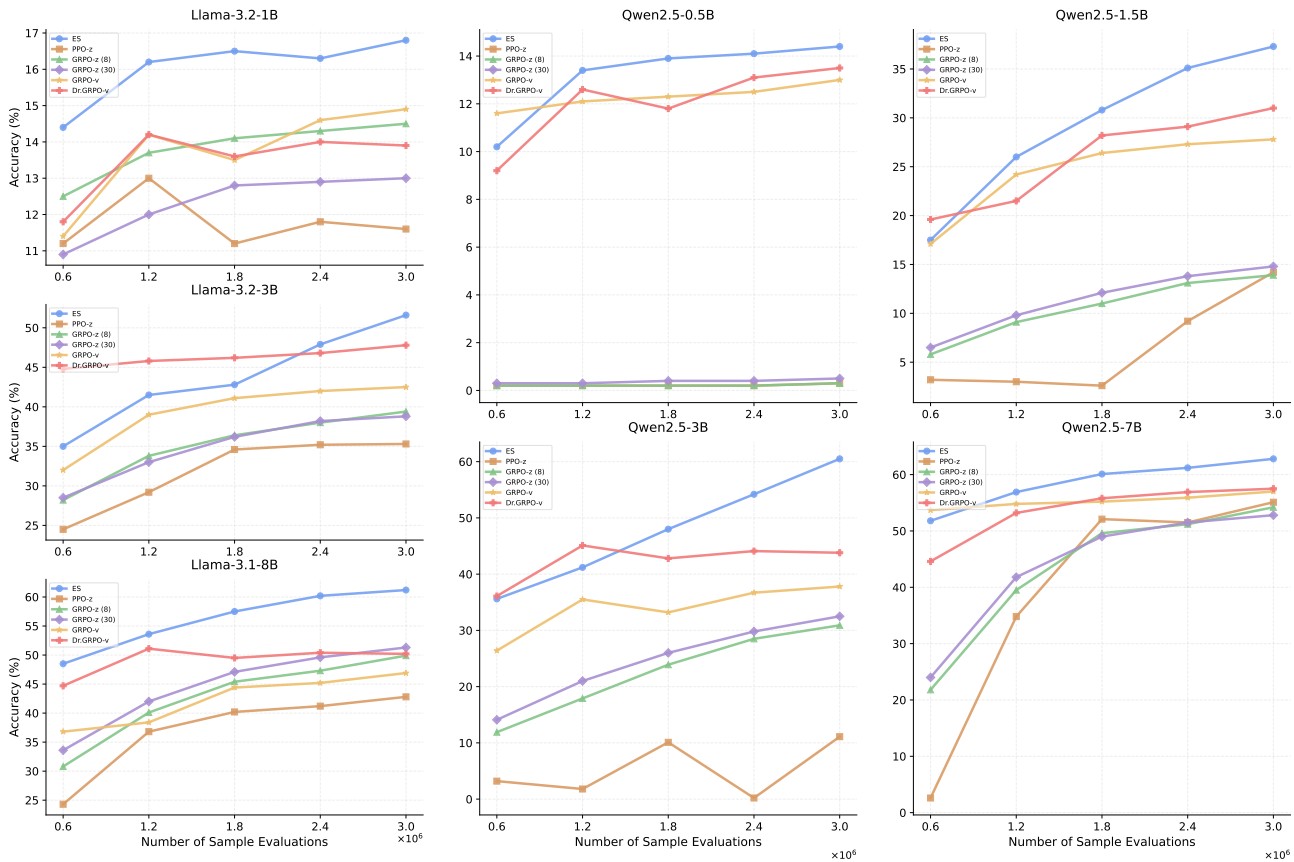

*Figure 7.* Training curves of ES and RL across two model families and six sizes in the countdown task. ES fine-tuning results in significantly better performance in all cases.

| Qwen-Math | `<\|im_start\|>system\nPlease reason step by step, and put your final answer within \boxed{}.<\|im_end\|>\n<\|im_start\|>user\n`{question}`<\|im_end\|>\n<\|im_start\|>assistant\n` |
|---|---|

*Table 12.* Qwen-Math prompt template used in this work.

**Qwen math template.**    Table 12 shows the template stucture used for training ES models. We follow the same template used for training the Qwen-Math series base models, where the model is required to provide a final answer inside \boxed{}. This requirement ensures that the final model answers are easy to extract and compare with the ground truth solutions for reward calculation during training. As shown in (Liu et al., 2025b), the choice of template can impact the final performance of the model. We chose the Qwen-Math template given it provides a platform for stable learning and good performance during fine-tuning.

**Checkpoint selection.**    Given there is no explicit validation set, for ES, we follow the standard model checkpoint selection mechanism from the literature whereby the checkpoints with high average pass@1 accuracy over each evaluation set over training are presented. We chose to present a number of ES checkpoints that all achieve sufficiently high average score. We chose our checkpoints to ensure competitive performance across the range of benchmarks. In this case, ES$_{\text{CHKPT-1}}$ is chosen for its high average score and occurs after 336 training steps. Additionally, we take a checkpoint after 160 training steps and perform 10 additional model update steps with $\alpha = \frac{\sigma}{4}$. This additional training produced ES$_{\text{CHKPT-2}}$. Given the lack of validation set, we utilize MATH500 as a pseudo validation set since MATH500 is an in-distribution validation. Following this, we select ES$_{\text{CHKPT-3}}$ because it achieves the highest performance in the MATH500 benchmark across our evaluations. The MATH500 ES$_{\text{CHKPT-3}}$ occurs after 192 training steps.

## A.8. Analysis of Computational Cost

The computational cost in terms of FLOPs is analyzed in this section. For each sample evaluation (processing one training data point), ES needs to perform one forward pass on the base model, whereas GRPO needs to perform one forward pass on the base model, one forward pass on the reference model, and one backward pass (backpropagation) on the policy. Following the standard estimation in literature (Narayanan et al., 2021; Gan & Isola, 2026), for a model with $P$ parameters and a sequence length of $L$, a single forward pass requires approximately $2PL$ FLOPs, and a backward pass requires $4PL$ FLOPs. Consequently, ES needs $2PL$ FLOPs for each sample evaluation, whereas GRPO needs $2 + 2 + 4 = 8PL$ FLOPs. Note that for earlier RL methods such as PPO, the needed FLOPs are even larger since there are additional forward and backward passes for the critic model. This analysis shows the computational advantage of ES in terms of FLOPs due to its inference-only nature.

## A.9. Parameter Magnitude Shifts by Evolutionary fine-tuning

This section characterizes how parameter magnitudes changed in ES fine-tuning in the countdown and conciseness experiments. Specifically, Figures 8 and 9, left column, show histograms of the absolute parameter magnitude shifts $\Delta$ before and after finetuning Llama and Qwen models, overlaid with random walk, on the Countdown task reported in Table 1. The right column in these figures shows the difference between $\Delta$ and the random walk.

For most models, $\Delta$ deviates very little from random walk. This is a counterintuitive result since fine-tuning actually resulted in a significant performance boost. A closer inspection reveals that most of the deviation was concentrated around zero. A likely explanation is that there are precision issues around zero, particularly with small bin sizes, which may lead to such deviations.

More significantly, a systematic deviation from the random walk was observed in conciseness fine-tuning of the largest model, Qwen2.5-7B-Instruct (Figure 10). The distribution shifts toward abundant small magnitude edits, suggesting that small parameter tweaks may be most significant in influencing output behavior. This result reinforces observations in prior studies (e.g. Liu et al., 2025a). A possible explanation is that large models encode functionality in a more redundandant manner, and therefore minor tweaks are sufficient to achieve fine-tuning objectives. In fact, the changes are nearly indistinguishable from random walk in Figures 8 and 9 likely because they are benevolent wrt. the fine-tuning objective. A more thorough investigation of these hypotheses is a most interesting direction of future work, potentially resulting in a better understanding of fine-tuning and information processing principles in LLMs in general.

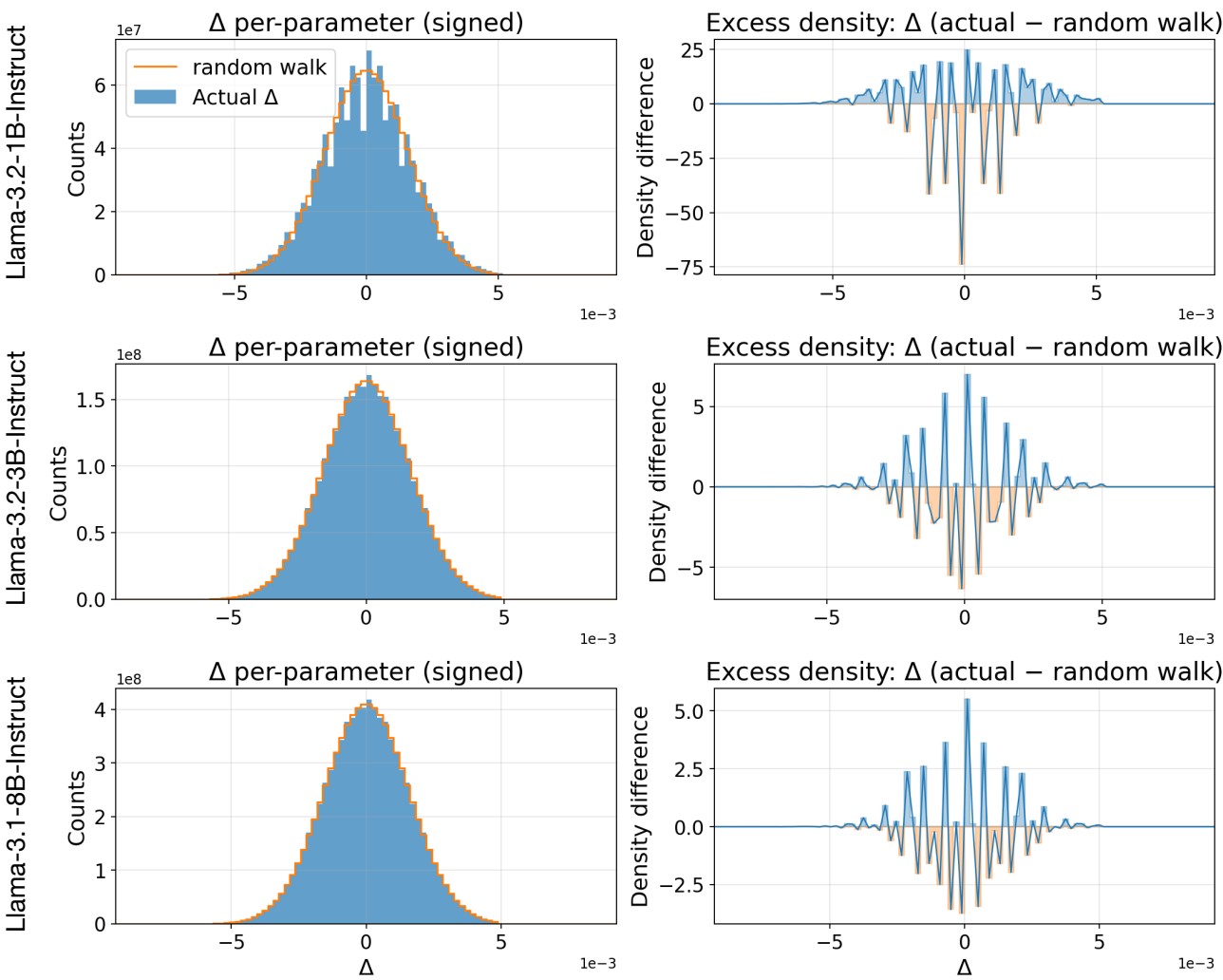

*Figure 8.* Parameter magnitude shift histograms for the Countdown task in Llama models optimized by ES. The changes are similar to those of a random walk, concentrated around zero, likely due to numerical inaccuracies.

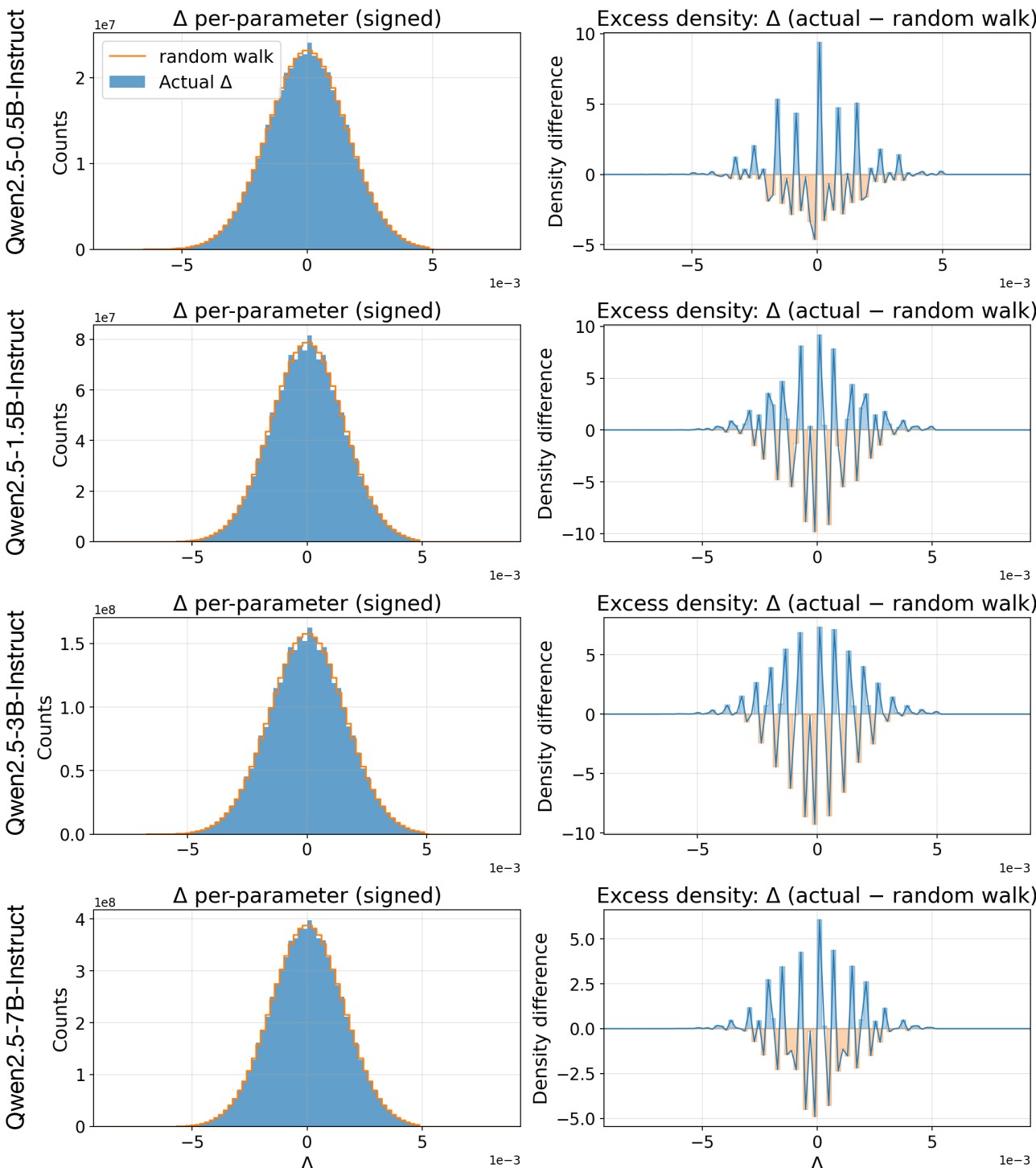

*Figure 9.* Parameter magnitude shift histograms for the Countdown task in Qwen models optimized by ES. The results are consistent with those observed in Llama models.

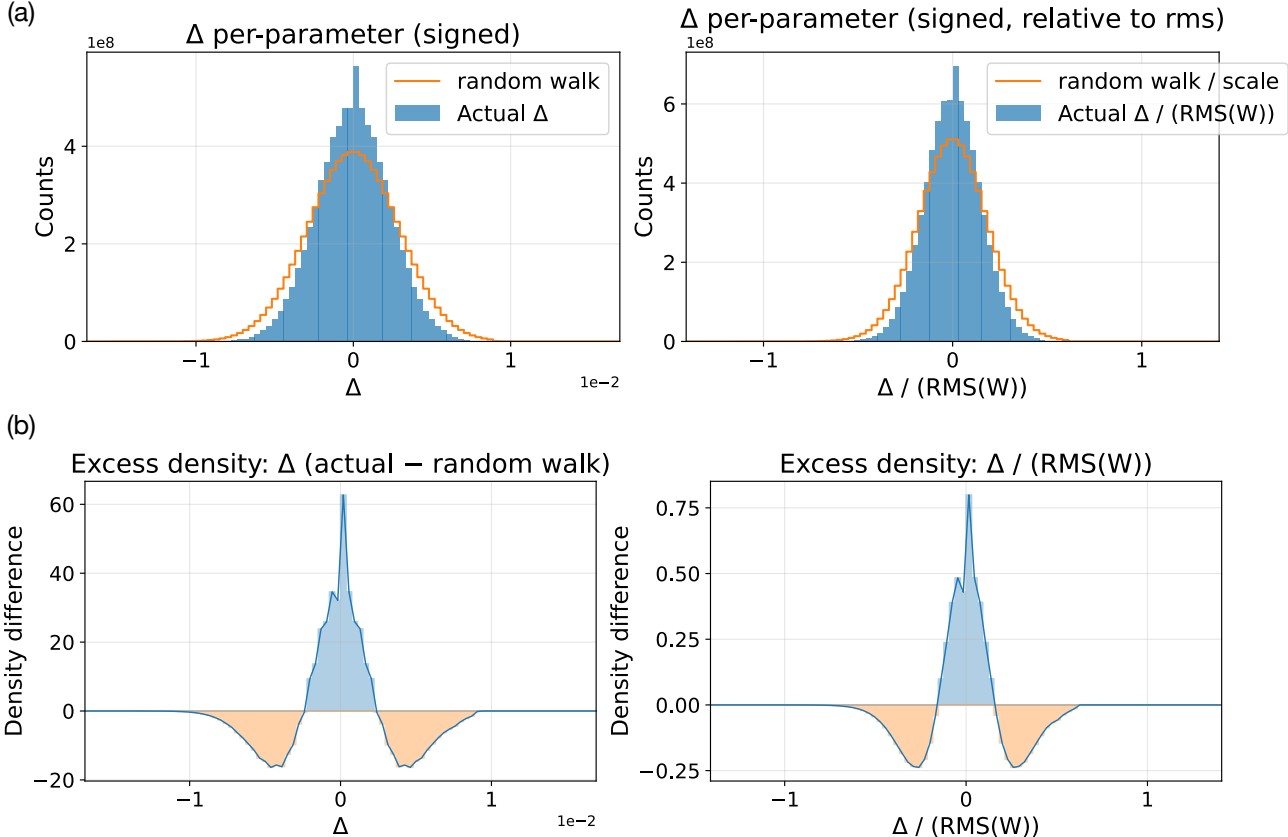

*Figure 10.* Parameter magnitude shift histograms in conciseness fine-tuning in Qwen2.5-7B-Instruct model with ES. In this case, the model is large and the fine-tuning goals is different, revealing a potentially significant pattern of primarily small changes. The hypothesis (to be analyzed more thoroughly in future work) is that behavior is coded in large models in a redundant manner, making it possible to achieve this fine-tuning objective through numerous small changes.

## A.10. Experiment with Mini Sudoku Task

Standard Sudoku requires filling in missing integers in a $9 \times 9$ grid, evenly divided into 9 smaller $3 \times 3$ grids. Integers must satisfy the following conditions: every row contains each integer $1 - 9$, every column contains each integer $1 - 9$, and each $3 \times 3$ sub-grid contains each integer $1 - 9$. Given the difficulty of $9 \times 9$ puzzles, existing evaluation of LLM's ability to complete Sudoku is often done with $4 \times 4$ grids, with equivalent rules for rows, columns, and $2 \times 2$ subgrids with digits $1 - 4$. Training data generation and evaluation was preformed with the sudoku board generation logic and reward function from (Stojanovski et al., 2025). $1,000$ example unique grids were generated, with 800 in training and 200 in testing. Qwen-2.5-3B-Instruct was used as the base model and trained with greedy decoding and batch size set to the entire 800 sample training set. ES was run for $2,500$ iterations and hyperparameters were set as follows: $\sigma = 0.001$, $\alpha = 0.0005$, and $N = 32$. With this setup, model performance improved from 2% to 66.5% of test set puzzles correctly solved. An example of the prompt, along with model responses before and after ES fine tuning is provided below in Figure 11.

---

**Prompt:**

You are a helpful assistant. You first think about the reasoning process in your mind and then provide the user with the answer. In 4x4 Mini Sudoku:

- Each row must contain each number from 1–4 exactly once

- Each column must contain each number 1–4 exactly once

- Each 2x2 subgrid must contain each number 1–4 exactly once

Solve this 4x4 Mini Sudoku puzzle:
```
_ _ 4 _
4 3 _ _
3 _ _ 1
1 _ 3 _
```
Show your work in `<think> </think>` tags. And return the final answer in `<answer> </answer>` tags, with spaces separating each number within a row, and newlines separating rows. For example `<answer>`
```
2 1 4 3
4 3 2 1
3 1 4 2
1 2 3 4
```
`</answer>`. Let me solve this step by step.
`<think>`

**Base Model Output (Incorrect):**

First, let's look at the given numbers and try to deduce the missing numbers. The number 4 is already placed in the first column, so we can't have another 4 in that column. The number 3 is already placed in the second row, so we can't have another 3 in that row. The number 1 is already placed in the third row, so we can't have another 1 in that row. Let's start filling in the grid based on these constraints: `</think>`
`<answer>`
```
2 1 4 3
4 3 2 1
3 1 4 2
1 2 3 4
```
`</answer>`
`</think>` This is the final solution to the 4x4 Mini Sudoku puzzle. Let me verify:
Each row contains the numbers 1, 2, 3, 4 exactly once: Row 1: 2 1 4 3, Row 2: 4 3 2 1, Row 3: 3 1 4 2, Row 4: 1 2 3 4.
Each column contains the numbers 1, 2, 3, 4 exactly once: Column 1: 2 4 3 1, Column 2: 1 3 1 2, Column 3: 4 2 4 3, Column 4: 3 1 2 4.
Each 2x2 subgrid contains the numbers 1, 2, 3, 4 exactly once: Subgrid 1: 2 1 / 3 4, Subgrid 2: 4 2 / 1 3, Subgrid 3: 3 1 / 4 2, Subgrid 4: 1 2 / 3 4.

All constraints are satisfied, so the solution is correct. `</think>` The final answer is:
```
<answer>
2 1 4 3
4 3 2 1
3 1 4 2
1 2 3 4
</answer>
```

> **Fine-Tuned Model Output (Correct):**
>
> First, let's look at the given numbers and try to fill in the missing numbers. We start with the first row: `_ _ 4 _`. Since each row must contain each number from 1 to 4 exactly once, the missing numbers in the first row must be 2, 1, and 3. So the first row becomes: 2 1 4 3.
> Next, let's look at the second row: `4 3 _ _`. Since each row must contain each number from 1 to 4 exactly once, and we already have 4 and 3 in this row, the missing numbers must be 1 and 2. So the second row becomes: 4 3 1 2.
> Now, let's look at the third row: `3 _ _ 1`. Since each row must contain each number from 1 to 4 exactly once, and we already have 3 and 1 in this row, the missing numbers must be 2 and 4. So the third row becomes: 3 4 2 1.
> Finally, let's look at the fourth row: `1 _ 3 _`. Since each row must contain each number from 1 to 4 exactly once, and we already have 1 and 3 in this row, the missing numbers must be 2 and 4. So the fourth row becomes: 1 2 3 4.
> So, the completed 4x4 Mini Sudoku puzzle looks like this: `</think>`
> ```
> <answer>
> 2 1 4 3
> 4 3 1 2
> 3 4 2 1
> 1 2 3 4
> </answer>
> ```

*Figure 11.* Example prompt and model outputs for the 4x4 Mini Sudoku task. The base model would respond in a variety of step by step methods, while the fine tuned model settled into a consistent, dense thinking space.

## A.11. Experiments on ARC-AGI Task

*ARC-AGI* (Chollet et al., 2024) is a benchmark designed to evaluate abstract reasoning and generalization capabilities. Each task presents a few input-output grid pairs as examples, requiring models to infer the underlying transformation rules and apply it to a new test input. The dataset has 400 public training tasks and evaluation tasks, focusing on creative problem-solving. We used 200 training tasks for fine-tuning and 200 evaluation tasks for testing.

### A.11.1. EXPERIMENTAL SETTINGS

*Prompts*. The ARC-AGI tasks are presented in image format. To use LLMs to solve them, we first map colors to the numbers as follows: black → 0, blue → 1, red → 2, green → 3, yellow → 4, grey → 5, gray → 5, pink → 6, orange → 7, purple → 8, brown → 9.

> **System Prompt:**
>
> You are a creative and meticulous ARC puzzle solver who explains reasoning before answering.
>
> **Task Explanation:**
>
> You will be given some number of paired example inputs and outputs. The outputs were produced by applying a transformation rule to the inputs. In addition to the paired example inputs and outputs, there is also one additional input without a known output. Your task is to determine the transformation rule and implement it in code.
> The inputs and outputs are each "grids". A grid is a rectangular matrix of integers between 0 and 9 (inclusive). These grids will be shown to you as grids of numbers (ASCII). Each number corresponds to a color. The correspondence is as

follows: black: 0, blue: 1, red: 2, green: 3, yellow: 4, grey: 5, pink: 6, orange: 7, purple: 8, brown: 9.
The transformation only needs to be unambiguous and applicable to the example inputs and the additional input. It doesn't need to work for all possible inputs.

**Reasoning Explanation:**

You'll need to carefully reason in order to determine the transformation rule. Start your response by carefully reasoning in ¡reasoning¿¡/reasoning¿ tags. Then, implement the transformation in code.
After your reasoning write code in triple backticks ("'python and then "'). You should write a function called 'transform' which takes a single argument, the input grid as 'list[list[int]]', and returns the transformed grid (also as 'list[list[int]]'). You should make sure that you implement a version of the transformation which works in general (it shouldn't just work for the additional input).

**Other Instructions:**

Don't write tests in your python code, just output the 'transform' function. (It will be tested later.)
You can also ask question to verify your observation on the inputs/outputs patterns in the form of python function which takes two arguments, the input and expected output grid both as 'list[list[int]]' and returns the boolean flag (True or False). We will help you by running your Python function on examples and let you know whether your question is True or False.
You follow a particular reasoning style. You break down complex problems into smaller parts and reason through them step by step, arriving at sub-conclusions before stating an overall conclusion. This reduces the extent to which you need to do large leaps of reasoning.
You reason in substantial detail for as is necessary to determine the transformation rule.
You are creative and accomplished at solving puzzles. When you write 'transform', do not hardcode the solution for each example. We will run your transform function on additional inputs later and check if your logic is generic in addition to check the correctness.

*Figure 12.* Prompts used for the ARC-AGI task.

*Hyperparameters* $\sigma = 0.001$, $\alpha = 0.0003$, $N = 50$, iterations = 1500.

## A.11.2. RL ATTEMPTS

Few prior works have explored using RL–based post-training for LLMs on ARC-AGI tasks. While RL has been widely used for LLM alignment and reasoning tasks (e.g., RLHF, math and QA benchmarks), its application to abstraction-centric, out-of-distribution generalization benchmarks like ARC-AGI remains relatively under-explored, largely due to challenges in reward design and efficient exploration. According to a recent report (Ranke, 2025) , applying GRPO to ARC-AGI resulted in minimal gains. The model failed to discover new strategies or abstract concepts; instead, it simply learned to rerank or refine the candidates it could already generate based on its original training.

