# OpenReview forum: "Evolution Strategies at Scale: LLM Fine-Tuning Beyond Reinforcement Learning"
_ICML.cc/2026/Conference — ICML 2026 regular_

### Official Review · Reviewer_syra · 2026-03-11

**Soundness:** 3
**Presentation:** 3
**Significance:** 2
**Originality:** 3
**Overall Recommendation:** 4
**Confidence:** 4

**Summary:**

This paper applies a variant of the evolution algorithm to fine-tuning LLMs. Using noise perturbation, the method achieves simultaneous inference of multiple policies required by the evolutionary algorithm, offering advantages in computational resources. The authors demonstrate that the method outperforms common RL algorithms such as GRPO and PPO on both countdown and math tasks.

**Compliance With Llm Reviewing Policy:**

Affirmed.

**Final Justification:**

The authors' responses during the rebuttal period have, to some extent, addressed the concerns regarding the soundness of the paper—specifically, through their explanations regarding stability and efficiency. However, I remain somewhat puzzled by the authors' explanation concerning the mathematical benchmarks, given that researchers have already developed more challenging benchmarks.
Still, I regard it as solid work. I hope the authors can incorporate what they justified in the rebuttal into the next version.

**Key Questions For Authors:**

1. Could the author discuss why the gains do not translate as effectively to broader reasoning tasks and whether ES faces unique bottlenecks when handling multi-step logic compared to structured games?
2.  Could the author include clearer comparisons in terms of training stability, sample efficiency, or computational cost?
3. Is the performance of ES sensitive to the perturbation scale?

**Limitations:**

The author has discussed the potential negative societal impact of their work. Suggestions and Limitations are presented in the weakness above.

**Strengths And Weaknesses:**

Strength:
1.   Scaling ES for LLM is theoretically important.
2.   Simple and gradient-free training pipeline: ES requires only forward inference and scalar rewards, which simplifies implementation and avoids the complexities of RL pipelines and gradient backpropagation.
3. Compared to GRPO, it can achieve higher rewards with smaller kl perturbations.


Weaknesses:
1.  The method shows clear improvements on the Countdown task, but the gains on mathematical reasoning benchmarks appear limited. This raises questions about the generality of the proposed approach and whether the benefits of ES extend beyond specific structured tasks.

2.  The introduction highlights several limitations of RL-based fine-tuning (e.g., instability and computational overhead). However, the paper does not provide sufficient empirical or analytical evidence showing that ES concretely addresses these issues. It would be helpful to include clearer comparisons in terms of training stability, sample efficiency, or computational cost.

3.  The paper mentions ES only requires response-level rewards. However, many recent RL-based approaches (e.g., RL with outcome rewards / RLVR-style methods) also operate with response-level feedback.

---

> ### Author Rebuttal · Authors · 2026-03-31
>
> We will address the weaknesses and questions identified in the review one by one.
>
> 1. W1/Q1. The math reasoning benchmarks have received much attention in the RL literature [1][2][3]. Performance in them is nearing a ceiling, and consequently, the remaining small differences are largely due to dedicated implementations. Nevertheless, even though the ES method was not subject to extensive hyperparameter tuning and sophisticated enhancements for these benchmarks, it is able to match the best performances reported in the literature (we picked the best-performing results/checkpoints from the RL literature). These results demonstrate the potential of ES in improving the state-of-the-art.
>
> To further evaluate the generality of ES, we ran comparisons on two other benchmarks: one in creative writing (ROCStories), and another in chemical reaction prediction  (USPTO):
>
> Results for Qwen2.5-1.5B-Instruct:
> |Qwen2.5-1.5B-Instruct|ES|GRPO|PPO|
> |-|-|-|-|
> |ROCStories|60.2|53.6|51.8|
> |USPTO|32.0|31.8|31.9|
>
> Results for Qwen2.5-3B-Instruct:
> |Qwen2.5-3B-Instruct|ES|GRPO|PPO|
> |-|-|-|-|
> |ROCStories|64.5|56.3|49.0|
> |USPTO|52.9|49.7|44.7|
>
> ES shows consistently better performance, demonstrating robustness and generality to different types of tasks.
>
> 2. W2/Q2. First, there are three aspects of training stability: insensitivity to hyperparameter setup, steady learning curves, and consistent performances across runs.
>
> Hyperparameter sensitivity: We performed extensive hyperparameter tuning for RL methods (Table 4 in the appendix, Section A.4 in the appendix, Figure 1 and Table 2), while only using a single fixed set of hyperparameters for ES. Still, ES exhibited consistently strong performance. To further evaluate the hyperparameter sensitivity of ES, we added a sweep experiment with varying $\sigma$ and $\alpha$ values (see Response W2 to Reviewer 8oKX). From the results, the performance of ES does not change significantly with different combinations of hyperparameters.
>
> Learning curves: Learning curves for ES and RL methods are included in Figure 7 in the appendix. ES generally shows more steady learning compared to RL.
>
> Performance across runs: Comparisons are included in Table 2 for the conciseness task; ES exhibits significantly lower variance. Similar results are seen in a new, similar experiment in the countdown task (see point 2 in the Response to Reviewer 2oyb).
>
> Second, regarding computational overhead/sample efficiency/computational cost, we included learning curves with regards to the total number of sample evaluations in Figure 7 of the appendix. Moreover, we added new analysis comparing ES and RL in terms of FLOPs and wall-clock time (GPU hours)--see Response W3 to Reviewer 8oKX. ES is computationally more efficient than RL in both FLOPs and wall-clock time.
>
> We will include the above considerations in the revision. Thanks for the suggestion.
>
> 3. W3. To clarify our point: Although existing RL fine-tuning methods work with outcome-only rewards, they are not as natural a fit for such a setting as ES is. In principle, RL requires step rewards at the token level to learn—however, such rewards are usually unavailable and difficult to assign [4][5]. Existing RL methods handle this issue by simply assigning the same reward to each token, which is not what they were designed for. In contrast, ES is designed to work inherently with response-level rewards, making it a more natural fit to outcome-based LLM fine-tuning tasks.
>
> 4. Q3. According to our new hyperparameter sweep experiments , ES is not very sensitive to the perturbation scale, as long as it is within a normal range; see Response W2 to Reviewer 8oKX for details.
>
> [1],[2],[3]: See response to Reviewer 2oyb.
>
> [4] “Do we need to verify step by step? Rethinking process supervision from a theoretical perspective”, ICML, 2025.
>
> [5] “The lessons of developing process reward models in mathematical reasoning”, ACL, 2025.
>
> [6] “Solving math word problems with process- and outcome-based feedback”, arXiv, 2022.

---

> > ### Author Rebuttal · Reviewer_syra · 2026-04-02
> >
> > Thank you to the author for the detailed response; my issue has been resolved, and I have decided to raise my rating.

---

> > > ### Author Response · Authors · 2026-04-02
> > >
> > > Thank you for spending time reading our paper and the rebuttal. Your suggestions indeed made our paper better.

---

### Official Review · Reviewer_QJMr · 2026-03-12

**Soundness:** 2
**Presentation:** 2
**Significance:** 3
**Originality:** 4
**Overall Recommendation:** 4
**Confidence:** 2

**Summary:**

This paper challenges the long-standing assumption that Evolution Strategies (ES) cannot scale to billion-parameter language models by demonstrating the first successful application of ES to full-parameter fine-tuning of LLMs without dimensionality reduction. The authors propose a memory-efficient ES implementation featuring layer-wise in-place perturbation, random seed-based noise retrieval, and decomposed parameter updates, enabling direct optimization in multi-B-parameter spaces with a surprisingly small population size (N=30). Extensive experiments across reasoning benchmarks (Countdown, math tasks), behavioral fine-tuning (conciseness), and challenging puzzles (ARC-AGI, Sudoku) show that ES can match or exceed RL-based methods (PPO, GRPO) while exhibiting advantages in reward-hacking resistance, cross-model robustness, and training stability. The work positions ES as a viable, backpropagation-free alternative paradigm for LLM post-training.

**Compliance With Llm Reviewing Policy:**

Affirmed.

**Final Justification:**

I would like to maintain the score of 3. I have some further questions to be answered.

1 Computational efficiency vs. methodological choice: If the computational cost of the proposed method is comparable to GRPO, what advantages does it offer over established GRPO-like approaches? Prior work has highlighted that SFT may lead to entropy collapse or overfitting in low-data regimes, potentially compromising generalization. Could the authors provide empirical entropy analysis or theoretical justification to demonstrate that their method mitigates these risks?

2. Broader model evaluation: In my initial review, I requested experiments on Qwen3/3.5 and Olmo3, which remain unaddressed. Given that Qwen3/3.5 generates substantially longer reasoning traces than Qwen2.5, evaluating the method's scalability and robustness on these newer architectures would significantly strengthen the empirical claims.

3. Comparison with improved GRPO variants: The current comparison focuses on the vanilla GRPO baseline. However, several recent works have identified theoretical limitations in the original GRPO formulation and proposed refined variants (e.g., GSPO). Including comparisons with such improved baselines would help better position the contributions of this work.

**Key Questions For Authors:**

1.  **Compute Efficiency at Scale:** Could the authors provide a detailed breakdown of FLOPs and GPU-hours for ES versus RL baselines, accounting for the cost of evaluating multiple perturbed models per iteration? How does the wall-clock training time compare when using the same hardware configuration?
2.  **Applicability to Learned Reward Models:** How would ES perform when the reward signal comes from a learned reward model (as in RLHF) rather than a rule-based verifier? Would the noise in reward model predictions exacerbate ES's sample inefficiency, or could ES's robustness to sparse rewards provide an advantage?
3.  **Decoding Strategy Sensitivity:** The experiments use greedy decoding for deterministic reward evaluation. How sensitive are the results to decoding strategy (e.g., sampling, beam search)? Would stochastic decoding during reward evaluation introduce additional variance that harms ES convergence?
4.  **Architecture Generalization:** The memory-efficient implementation relies on uniform layer-wise perturbation. How would the approach scale to models with heterogeneous architectures (e.g., Mixture-of-Experts, multi-modal encoders) where parameters have different roles or sparsity patterns?
5.  **Interpretability of Parameter Changes:** The paper observes that parameter magnitude shifts resemble random walk. What does this imply about the nature of fine-tuning in high-dimensional spaces? Are there more efficient ways to guide ES updates using task-specific inductive biases (e.g., structured perturbations, layer-wise adaptive noise)?

**Limitations:**

Yes.

**Strengths And Weaknesses:**

## Strengths

1.  **Conceptual Breakthrough in Scaling Zeroth-Order Optimization:** The paper overturns a widely held belief in the community by providing empirical evidence that ES can effectively search billion-parameter spaces. This opens a new design space for gradient-free post-training methods, particularly valuable for settings where gradient computation is expensive or unreliable.
2.  **Practical Engineering Innovations for Memory Efficiency:** The implementation details—layer-wise in-place perturbation, seed-based noise reconstruction, and decomposed updates—are clever and well-executed. These techniques significantly reduce peak GPU memory usage, making full-parameter ES feasible on commodity hardware and democratizing access to alternative fine-tuning paradigms.
3.  **Demonstrated Behavioral Advantages over RL:** The empirical findings that ES exhibits reduced reward hacking, more consistent results across random seeds, and better robustness to base model choice are compelling. The Pareto-front analysis in the conciseness task provides strong evidence that ES discovers qualitatively different (and often preferable) optimization trajectories compared to gradient-based RL.

## Weaknesses

1.  **Sample Efficiency and Compute Cost Analysis is Incomplete:** While the paper emphasizes a small population size (N=30), it does not fully account for the computational cost of evaluating 30 perturbed models per iteration. Each evaluation requires a full forward pass through a billion-parameter model, and the total number of forward passes per ES iteration is N×(batch size). In contrast, RL methods can leverage batched inference, gradient accumulation, and asynchronous actor-learner architectures to amortize compute. The paper's "same total sample evaluations" comparison may obscure significant differences in wall-clock time and FLOPs. A more rigorous efficiency analysis—including FLOP counts, GPU-hours, and scalability to larger populations—would strengthen the claims about practical viability.
2.  **Experimental Scope Limited to Verifiable, Outcome-Only Tasks:** All benchmarks use rule-based, binary rewards (correct/incorrect) or simple length-based penalties. These settings favor ES's strength in outcome-only optimization but do not test its applicability to more complex fine-tuning scenarios: (a) tasks with learned reward models (e.g., RLHF with preference data), where reward signals are noisy and potentially misaligned; (b) open-ended generative tasks (dialogue, creative writing) where reward design is ambiguous; (c) multi-objective alignment requiring trade-offs between helpfulness, harmlessness, and honesty. The generalization of ES to these realistic post-training settings remains an open question.
3.  **Gradient-Free Optimization May Limit Fine-Grained Control:** By operating without backpropagation, ES cannot leverage token-level or intermediate-representation signals that gradient-based methods exploit. For tasks requiring precise behavioral control (e.g., factual grounding, style transfer, safety constraints), the inability to assign credit to specific model components or generation steps may limit ES's effectiveness. The paper does not explore whether ES can be combined with auxiliary gradient signals or whether hybrid approaches might offer better trade-offs.
4.  **Hyperparameter Sensitivity and Lack of Adaptive Mechanisms:** The paper uses fixed hyperparameters (σ=0.001, α=5×10⁻⁴) across all experiments, claiming robustness. However, the conciseness experiments reveal that σ still requires tuning to balance reward optimization and KL divergence. More critically, the noise scale σ and learning rate α likely interact with model size, task difficulty, and reward sparsity in non-trivial ways. The paper does not provide guidance on adapting these hyperparameters automatically (e.g., via adaptive ES variants or meta-learning), which would be essential for broader adoption.
5.  The observation that N=30 works for billion-parameter models is counterintuitive given classical ES theory, where sample complexity typically scales with parameter dimension. The paper hypothesizes connections to low intrinsic dimensionality of LLM loss landscapes but provides no empirical validation (e.g., via effective rank analysis or Hessian spectra). Without understanding the conditions under which small populations suffice, it is difficult to predict when ES will fail or to design more efficient variants. A deeper analysis of the optimization dynamics—perhaps via landscape visualization or convergence diagnostics—would strengthen the methodological contribution.
6. How about most adavanced open-source models like Qwen3/3.5 and Olmo3? Please add new experiments on these advanced models to make the experiments more principled and convinced.

---

> ### Author Rebuttal · Authors · 2026-03-31
>
> We will address the weaknesses and questions one by one.
>
> W1/Q1. Actually our claim is that ES is more computationally efficient than RL given the same total number of sample evaluations. As evidence, we ran a new analysis comparing ES and RL in terms of both FLOPs and wall-clock time (GPU hours in the same hardware). Please see Response W3 to Reviewer 8oKX for details. Without backpropagation, ES has significantly lower computational cost in both dimensions. Moreover, further experiment shows that small populations are sufficient; i.e. the benefit from larger populations is small (see point 6 in the Response to Reviewer 2oyb).
>
> W2. The selection of the benchmarks follows the main-stream RL fine-tuning literature [1][2][3], in which rule-based verifiable problems are used. Thus, ES is evaluated in the same tasks where SOTA RL were demonstrated, instead of picking problems that would favor ES. The reason why these benchmarks are standard is that their evaluations are clean and deterministic, allowing a fair comparison of different fine-tuning methods without unmeasurable influence from a noisy reward model. Moreover, these math reasoning benchmarks used are very challenging, including competition-level mathematical problems that require advanced multi-step reasoning.
>
> It is, however, interesting to evaluate ES in a broader set of benchmarks as well. To this end, we ran new experiments in a creative writing benchmark (ROCStories) and in a chemical reaction prediction benchmark (USPTO). ES performs well on both; see Response W1/Q1 to Reviewer syra for details.
>
> W3. Even though in principle gradient-based methods can learn from step rewards, proper credit assignment at token level is difficult and possibly unhelpful [4][5][6]. As a result, SOTA RL methods mostly treat them as outcome-only scenarios [1][2][3].  ES is actually a more natural fit for such scenarios (as detailed in Response W3 to Reviewer syra), and therefore a viable alternative.
>
> Combining ES with RL may indeed be feasible: For example, it may be possible to use ES for global exploration and RL for fine-grained exploitation. Such synergies are an interesting direction of future work.
>
> W4. To further evaluate the parametric sensitivity of ES, we ran new experiments for ES with different combinations of $\sigma$ and $\alpha$ (see Response W2 to Reviewer 8oKX for details). Its performance does not change significantly as long as $\sigma$ and $\alpha$ are within normal ranges.
>
> In order to adapt hyperparameters automatically, one potential way is to measure the performance variance of the perturbations in each iteration, then adjust $\sigma$ accordingly to maintain a reasonable variance, and adapt $\alpha$ proportionally with $\sigma$. It will be an interesting opportunity–thanks for suggesting it.
>
> W5. We empirically validated the low-dimensionality of the reward landscape with Hessian-spectrum-derived metrics of the ES-smoothed reward objective, using a perturbation-based Hessian–vector product estimator (no autograd) and stochastic Lanczos quadrature (SLQ). The results show that LLMs with 0.5B-7B parameters only have an effective rank of hundreds, and there is an increasingly smaller effective rank given a larger model size. This result significantly improves our understanding of why ES works, and will be an important addition to the paper.
>
> W6. As suggested, we ran an experimental comparison on Olmo3 as well:
> |Olmo3-7B-Instruct|ES|GRPO|PPO|
> |-|-|-|-|
> |Countdown|71.0|68.5|69.0|
> |Math500|69.9|63.5|63.1|
> |OlympiadBench|33.1|27.9|28.0|
>
> Thus, ES performs consistently better on Olmo3 as well, demonstrating that it is robust across model types.
>
> Q2. As a population-based method, ES should be more robust to noise in reward models than a single policy learner. Aggregating learning signals from multiple perturbed models should mitigate the influence of a spiking noise. Testing these hypotheses is an interesting direction of future work.
>
> Q3. As suggested, we ran an ablation study using different decoding temperatures during ES training (Qwen2.5-3B-Instruct on Countdown):
>
> |Temperature|0|0.5|1.0|
> |-|-|-|-|
> |Accuracy|59.4|62.5|59.4|
>
> Thus, using a temperature larger than 0 does not harm ES convergence. Instead, a moderate temperature may even improve it.
>
> Q4.  Actually the memory-efficient implementation does not rely on uniform layer structure. It can be adjusted to any block size (not necessarily layer-level), e.g., each block can be a subset of parameters of any size. As a result, it can be readily applied to any architecture design.
>
> Q5. The apparent random walk behavior indicates that most parameter changes do not have a critical effect on the reward. However, this is also an opportunity: The search may be more efficient if we can identify these non-critical parameters and focus on others. This will be an important direction of future work.
>
>
> [1],[2],[3]: See response to Reviewer 2oyb
>
> [4],[5],[6]: See response to Reviewer syra

---

> > ### Author Rebuttal · Reviewer_QJMr · 2026-04-03
> >
> > Thank you for your detailed rebuttal. I appreciate the authors' efforts in addressing my concerns, and I have a few follow-up questions for further clarification:
> >
> > 1. Computational efficiency vs. methodological choice: If the computational cost of the proposed method is comparable to GRPO, what advantages does it offer over established GRPO-like approaches? Prior work has highlighted that SFT may lead to entropy collapse or overfitting in low-data regimes, potentially compromising generalization. Could the authors provide empirical entropy analysis or theoretical justification to demonstrate that their method mitigates these risks?
> >
> > 2. Broader model evaluation: In my initial review, I requested experiments on Qwen3/3.5 and Olmo3, which remain unaddressed. Given that Qwen3/3.5 generates substantially longer reasoning traces than Qwen2.5, evaluating the method's scalability and robustness on these newer architectures would significantly strengthen the empirical claims.
> >
> > 3. Comparison with improved GRPO variants: The current comparison focuses on the vanilla GRPO baseline. However, several recent works have identified theoretical limitations in the original GRPO formulation and proposed refined variants (e.g., GSPO). Including comparisons with such improved baselines would help better position the contributions of this work.

---

> > > ### Author Response · Authors · 2026-04-05
> > >
> > > Thank you for your additional questions. We will address them one by one below:
> > >
> > > Q1. We want to clarify that our analysis on computation time and FLOPs (see Response W3 to Reviewer 8oKX for details) indicates that ES has more computational benefits than GRPO since ES completely gets rid of backpropagation calculations. Beyond computational efficiency, ES has several other advantages: (a) As a backpropagation-free method, the peak GPU memory is much lower than that of GRPO which requires activation values, gradients and optimizer states to be cached and stored. This means that ES can be run on more limited hardware. (b) Without the gradient calculations, engineering overhead/complexity is significantly reduced, making ES easier to parallelize in large-scale training. Please see paragraph “Engineering benefits of ES vs. RL” in Section 5 of the original paper for more details on these two points. (c) ES is less sensitive to hyperparameter setup, as demonstrated in the experimental setup in the paper:  we did an extensive hyperparameter tuning for GRPO on each task, while only using the same fixed set of hyperparameters for ES across all experiments. Our original rebuttal to  W4 also included more parametric sensitivity analysis for ES. This advantage of ES can reduce the expensive overhead usually incurred in tuning RL hyperparameters. (d) ES is more stable across runs, as shown in Table 2 in the paper and Response 2 to Reviewer 2oyb. This stability can save the expected cost in practice to obtain a reliable fine-tuning result. (e) ES has a lower tendency to reward hacking, as shown in Section 4.2 and discussed in Section 5 of the paper. We also added further clarifications in Response 4 to Reviewer 2oyb.
> > >
> > > Second, as suggested, we ran experiments on entropy analysis for ES in a low-data regime (512 training data sampled from GSM8K). For each prompt, 32 responses were sampled with a temperature of 0.6. The entropy for the top 20 tokens at each token position was then calculated and averaged for each response. Finally, the entropy values were binned, and the frequencies for each bin were reported below for the model before and after ES fine-tuning:
> > >
> > > Results for Qwen2.5-1.5B-Instruct:
> > > |Entropy Bin|[0,0.5)|[0.5,1)|[1,1.5)|[1.5,2)|[2,2.5)|[2.5,3)|[3,3.5)|[3.5,4)|
> > > |-|-|-|-|-|-|-|-|-|
> > > |Frequency for the original model|7360|497|348|220|132|55|30|4|
> > > |Frequency for ES fine-tuned model|6568|973|842|478|278|158|53|19|
> > >
> > > Results for Qwen2.5-3B-Instruct:
> > > |Entropy Bin|[0,0.5)|[0.5,1)|[1,1.5)|[1.5,2)|[2,2.5)|[2.5,3)|[3,3.5)|[3.5,4)|
> > > |-|-|-|-|-|-|-|-|-|
> > > |Frequency for the original model|5947|1069|972|563|257|104|45|18|
> > > |Frequency for ES fine-tuned model |5996|1106|858|524|181|67|13|3|
> > >
> > > Thus, ES fine-tuning slightly increased entropy in general for the 1.5B model (i.e. reduced the number of low entropy cases and increased the number of high entropy cases) while almost maintaining the entropy distribution for the 3B model. No entropy collapse was observed.
> > >
> > > To further validate this conclusion, we ran an experiment to evaluate the pass@k accuracies under different k values for models before and after ES fine-tuning:
> > >
> > > Results for Qwen2.5-1.5B-Instruct:
> > > |k|1|2|3|4|8|16|32|
> > > |-|-|-|-|-|-|-|-|
> > > |pass@k accuracy for the original model|0.250|0.396|0.491|0.557|0.699|0.805|0.881|
> > > |pass@k accuracy for ES fine-tuned model|0.441|0.619|0.709|0.764|0.862|0.920|0.951|
> > >
> > > Results for Qwen2.5-3B-Instruct:
> > > |k|1|2|3|4|8|16|32|
> > > |-|-|-|-|-|-|-|-|
> > > |pass@k accuracy for the original model|0.539|0.683|0.745|0.782|0.854|0.911|0.954|
> > > |pass@k accuracy for ES fine-tuned model|0.790|0.870|0.900|0.916|0.944|0.962|0.972|
> > >
> > > Thus, ES consistently improved the pass@k accuracies with different k values, without entropy  collapse. Therefore, loss of generalization in low-data regimes is unlikely with ES.
> > >
> > > Q2. Following your suggestion, we ran experiments on Olmo 3; see response W3 in the initial rebuttal. These experiments respond to your concern in two ways: (a) Olmo 3 is from a different model family from Qwen, thus broadening the coverage of  the experiments. (b) Based on Olmo 3 official report [1], Olmo 3 also benefits from its long thinking process and performs competitively compared to Qwen 3 in many benchmarks (Qwen 3.5 was released after ICML submission deadline and not sufficiently investigated in the literature at this point). Regarding the choice of Qwen 2.5 for most experiments: Qwen 2.5 models are the most commonly used  models in RL fine-tuning literature, providing comprehensive comparisons with RL state-of-the-art.
> > >
> > > Q3. Actually the comparisons were not based on vanilla GRPO; in each domain, the state-of-the-art variant for that domain was used, including Dr. GRPO. In contrast, we kept ES simple without sophisticated enhancements. Our motivation was to demonstrate that a simple but fundamentally different approach compares well with state-of-the-art RL.
> > >
> > > [1] Olmo 3 official model card, https://huggingface.co/allenai/Olmo-3-7B-Think .

---

### Official Review · Reviewer_2oyb · 2026-03-12

**Soundness:** 3
**Presentation:** 3
**Significance:** 3
**Originality:** 3
**Overall Recommendation:** 5
**Confidence:** 3

**Summary:**

The paper studies the use of Evolutionary Strategies (ES) as an alternative to Reinforcement Learning (RL) for the finetuning of Large Language Models (LLMs). The authors present a stripped-down evolutionary strategies algorithm and evaluate it on multiple kind of tasks:
 - the countdown task, a task where RL finetuning have been previously been reported to suffer generalization issues
 - a question answering task focused on the conciseness metric
 - math reasoning tasks
 - puzzle tasks (ARC-AGI, Sudoku)


The paper provides a thorough comparison with RL methods on the countdown and conciseness task, revealing superior performance of ES across the board. On the math reasoning tasks, ES are compared with multiple baselines and show overall a strong performance. On puzzle tasks, ES fine-tuning is presented as a proof of concept, showing it can drastically improve the performance over the base model (but no comparison with existing fine tuning methods).

In Section 5 the authors discuss the results obtained in the study and provide some insights and preliminary explanation to suggest why ES tend to not fall into the same pitfalls as RL finetuning (reward hacking, high variance across seeds, sensitivity to hyperparameters).

**Compliance With Llm Reviewing Policy:**

Affirmed.

**Final Justification:**

The paper is a clean, based experimental study about the use of ES for fine-tuning LLMs. My initial assessment was that the paper was relevant and deserved publication. After the rebuttal, the authors clarified a few points and improved the presentation of the results. I therefore maintain my score and recommendation (accept).

**Key Questions For Authors:**

1. Do you consider your characterization of reward hacking sufficient to support the claim that ES is less prone to it? In particular, reward hacking seems to have been identified qualitatively by inspection rather than through a systematic or quantitative measure.

 2. Table 2 seems to suggest that a higher noise scale introduces better rewards but also a higher KL divergence. Did you find that by increasing the noise scale further ES was more subject to reward hacking?

 3. Can you justify the experimental setting of the conciseness task regarding the number of prompts used?

 4. Can you provide or justify the absence of confidence measures ?

 4. Did you try some of the ES improvements you mention in the paper in some of your unreported experiments? It would interesting to discuss in the paper whether or not you think the gap between the ES you tried and the best ES you could have made is large.

**Limitations:**

I think the limitations of the work are quite well addressed.

**Strengths And Weaknesses:**

### Soundness

I found the work to be technically sound. I think the experiments are well designed and the results are compelling. I appreciate the fact authors focused on a minimal implementation of ES with a simple implementation. The memory management proposed by the authors (saving only random seeds instead of parameters) and updating parameters layer by layer in place is sound. This incorporates well into a broader discussion about memory tradeoffs that was well-covered by the authors, and typically important to address since population based methods tend to have high-memory footprint.

However, I think a deeper discussion on the computational costs in terms of number of forward pass to convergence or in terms of FLOPs used could strengthen the paper, and help better understand the tradeoffs between RL and ES.

### Presentation
The paper is overall easy to follow and pleasant to read. The results are well-presented and figures are informative. However, I couldn't find error bars or condidence intervals in multiple places such as Table 1 or Figure 2, nor a justification for the absence of it. I think this would improve the presentation as well as being more transparent about whether or not methods are statistically separable.


### Significance

I believe the paper has a good experimental depth, but I'd say authors can be more upfront about the choice of baselines (or the absence of baselines) for example on the puzzle tasks. While this is not a problem in itself, this could help the reader understand the choices made by the authors as well as the future potential work and directions to explore.

Something I found handwavy was the characterization of reward-hacking, I didn't find any quantitative measure of it, and it seems to have been identified at hand by the authors. This weakens the claim that ES methods are less prone to reward hacking.

Additionnaly I was surprise to see that the conciseness fine-tuning uses only 2 training prompts as shown in Table 5. And only a few evaluations prompt. I don't know very well this task but I found it surprising.


The paper highlights the population size N=30 as surprisingly small, but provides no ablation over N. Given that this is one of the most counterintuitive claims of the paper, an ablation (or at least a discussion of sensitivity) would be valuable.

### Originality

The work provides a good demonstration of the use of ES for fine-tuning complete LLMs. To the best of my knowledge this is the first study of this scale, using Billion-scale LLMs and using full fine-tuning instead of parameter efficient fine-tuning.

---

> ### Author Rebuttal · Authors · 2026-03-31
>
> We will address the weaknesses and questions identified in the review one by one.
>
> 1. Computational cost: We ran a computational cost analysis in terms of both FLOPs and wall-clock time; see Response W3 to Reviewer 8oKX for details.
>
> 2. Statistical significance and Q4: For the experiments in Table 1 and Figure 2, we follow the standard setup in SOTA RL literature [1][2][3], i.e., one run for each training configuration, due to the high training cost of LLMs. However, the conciseness task is cheaper, and we were able to perform several runs. Table 2 shows the results with mean and standard deviation.
>
> For this rebuttal, we also extended the countdown experiment with three runs for each configuration:
> | Mean+Std|ES|GRPO-v|DR.GRPO-v|
> |-|-|-|-|
> |Qwen2.5-1.5B-Instruct|$37.73\pm\mathbf{1.14}$|$26.77\pm1.98$|$32.97\pm2.05$|
> |Qwen2.5-3B-Instruct|$58.67\pm\mathbf{1.47}$|$35.07\pm2.33$|$44.43\pm1.89$|
>
> ES generally has a lower variance across runs, and the performance differences between ES and other RL methods are statistically significant with p < 0.05 using Student’s $t$-test. We will add the clarifications and results.
>
> 3. Baselines: Good point; to clarify, for the countdown task, the baselines cover a wide range of implementation variants of SOTA RL methods. For the conciseness task, GRPO was implemented based on an SOTA RL library (trl). For the math reasoning benchmarks, the best-performing results/checkpoints reported in SOTA RL literature were selected. For the puzzle problems, there are no well-established RL fine-tuning baselines with which to compare (as discussed in Section A.9.2 in the appendix), but we believe they are valuable to demonstrate the breadth of the approach.
>
> 4. Reward hacking and Q1: Reward hacking was actually detected quantitatively:  the KL divergence of each fine-tuned model with regard to the original base model was calculated, and a model was automatically marked as a “possible anomaly” if its KL divergence exploded and exceeded a detection threshold. Only for these “possible anomaly” models, the response details were checked to confirm the reward hacking behavior. This automatic process is fair for both ES and RL. Indeed, a strong correlation between reward hacking and KL divergence was observed, and it was significantly more frequent for RL runs. We will clarify this point in the revision.
>
> 5. Number of training samples and Q3: There are two reasons for using a very small number of training samples for the conciseness experiments: (1) We need to make sure that the fine-tuning task is cheap to train, so that we can iterate over a large number of configurations and perform multiple runs for each configuration. In this manner, we can investigate the behavioral differences between ES and RL more comprehensively. (2) We found that only 2 training samples were indeed sufficient for ES and RL to learn this model-level objective, so further examples were simply not needed.
>
> 6. Population size: To further clarify the effect of population size N, we ran an ablation study with varying N on a small math reasoning dataset (512 prompts sampled from GSM8K and evaluating on the full test dataset of GSM8K):
> Results for Qwen2.5-1.5B-Instruct:
> |Population Size|16|30|64|128|
> |-|-|-|-|-|
> |Accuracy|0.697|0.714|0.725|0.727|
>
> Results for Qwen2.5-3B-Instruct:
> |Population Size|16|30|64|128|
> |-|-|-|-|-|
> |Accuracy|0.807|0.823|0.804|0.831|
>
> The conclusion is that a larger population size generally leads to a better performance, but the differences are small. In particular, N=30 is a good tradeoff of performance and effort.
>
> 7. Q2. Effect of noise: Indeed when we increased the noise to an extremely large value such as 0.004, we occasionally observed reward hacking with ES. However, within the standard noise range  of 0.0005 to 0.0015 we have never seen reward hacking behavior for ES.
>
> 8. Q5.Good question! We tried mirrored sampling in a small number of tests, and found it to improve the stability of learning curves, though not conclusive yet. Other experiments are ongoing: weight decay should further reduce the unnecessary weight divergence from the base model, virtual batch normalization should further improve the learning stability, and more advanced optimizers with adaptive learning rate should further improve the optimization performance. Overall, with these enhancements, the performance should improve further, and the approach should become more robust across tasks. These are all interesting directions for future work.
>
> [1] “DeepSeekMath: Pushing the Limits of Mathematical Reasoning in Open Language Models”, arXiv, 2024.
>
> [2] “Simplerl-Zoo: Investigating and taming zero reinforcement learning for open base models in the wild”, Conference on Language Modeling, 2025.
>
> [3] “Understanding R1-Zero-like training: A critical perspective”, Conference on Language Modeling, 2025.

---

> > ### Author Rebuttal · Reviewer_2oyb · 2026-04-03
> >
> > The authors answered adequately to my remarks and questions. I think the additional clarification will improve the paper's presentation. I therefore maintain my positive evaluation and my recommendation (accept).

---

> > > ### Author Response · Authors · 2026-04-03
> > >
> > > Many thanks for your thoughtful comments and constructive suggestions. Your input indeed helped us to further improve the paper quality. Thank you again for your time and effort in reviewing our work.

---

### Official Review · Reviewer_8oKX · 2026-03-13

**Soundness:** 3
**Presentation:** 3
**Significance:** 4
**Originality:** 3
**Overall Recommendation:** 5
**Confidence:** 4

**Summary:**

This paper presents a novel application of Evolution Strategies (ES) for full-parameter fine-tuning of billion-parameter Large Language Models (LLMs), challenging the dominance of Reinforcement Learning (RL) in LLM post-training. The authors demonstrate that ES, despite long-standing assumptions about its scalability limitations, outperforms RL across key metrics including tolerance to long-horizon rewards, robustness across base models, resistance to reward hacking, and training stability. The work introduces a memory-efficient ES implementation with parallelization support, validated through extensive experiments on reasoning benchmarks (Countdown, math tasks), conciseness fine-tuning, and puzzle-solving (Sudoku, ARC-AGI).

**Compliance With Llm Reviewing Policy:**

Affirmed.

**Final Justification:**

According to the discussion, I would keep my positive evaluation.

**Key Questions For Authors:**

1. Could ES be combined with RL (e.g., using ES for exploration and RL for exploitation) to further improve fine-tuning?
2. How to integrate ES into a complete model postpretraining pipeline?
3. For scaling further on larger models, what issues may occur and how to resolve them?

**Limitations:**

Yes

**Strengths And Weaknesses:**

### Strengths
1. The proposed evolutionary strategies for LLM finetuning breaks the critical barrier of scaling ES to billion-parameter LLMs without dimensionality reduction. It aims to challenge the assumption that ES is infeasible for large models, which is insightful for the community.
2. The use of a small population size (N=30) for high-dimensional optimization is counterintuitive and impactful.
3. It is well-structured with clear motivation, method details, and experimental validation. The comparison between ES and RL is systematic, and technical appendices provide sufficient implementation details.
4. The paper proposes a gradient-free alternative to RL, reducing engineering complexity and memory costs. This could democratize LLM fine-tuning and inspire new directions in optimization for large-scale models.
5. For the comparison of ES and RL The paper conduted comprehensive impirical studies, analyzing from multiple perspectives including training features, parameter distribution, and downstream tasks.

### Weaknesses
1. While the scalability of ES to LLMs is novel, the core ES algorithm (simplified NES with Gaussian noise) is not fundamentally new. The paper could better contextualize how its contributions advance beyond prior ES variants (e.g., OpenAI ES, CMA-ES).
2. The hyperparameter sensitivity analysis for ES is limited. For example, how do $\sigma$ and $\alpha$ affect performance across model sizes?
3. It lacks analysis or experiment  on the computational cost.

---

> ### Author Rebuttal · Authors · 2026-03-31
>
> We will address the weaknesses and questions one by one.
> 1. W1. The idea was to keep the method simple to demonstrate the advantages of ES fine-tuning without confounding factors. A few innovations were introduced to reduce the GPU memory requirement and computational cost in order to scale to billion-parameter models. Compared to both OpenAI ES and CMA-ES, the main differences include z-score normalization of rewards, layer-level in-place perturbation and restoration, and decomposition of the parameter update. In addition, the covariance calculation and adaptation of CMA-ES was completely omitted because it turned out infeasible in the billion-parameter space. We will clarify these innovations in the revision.
>
> 2. W2. As suggested, we ran a hyperparameter sweep experiment for ES with varying $\sigma$ and $\alpha$ values on a small math reasoning dataset (512 prompts sampled from GSM8K and evaluating on the full test dataset of GSM8K):
>
> Results for Qwen2.5-1.5B-Instruct:
> |$\sigma$|$\alpha$|Accuracy|
> |-|-|-|
> |0.001|0.0005|0.714|
> |0.002|0.001|0.688|
> |0.0005|0.00025|0.654|
> |0.002|0.0005|0.726|
> |0.0005|0.0005|0.675|
> |0.001|0.001|0.660|
> |0.001|0.00025|0.704|
>
> Results for Qwen2.5-3B-Instruct:
> | $\sigma$ | $\alpha$ | Accuracy |
> |-|-|-|
> |0.001|0.0005|0.817|
> |0.002|0.001|0.808|
> |0.0005|0.00025|0.792|
> |0.002|0.0005|0.826|
> |0.0005|0.0005|0.791|
> |0.001|0.001|0.782|
> |0.001|0.00025|0.823|
>
> These results demonstrate that the performance of ES is robust across a range of hyperparameter choices.
>
> 3. W3. As suggested, we analyzed computational cost in terms of both FLOPs and wall-clock time (GPU hours):
> For each sample evaluation (processing one training data point), ES needs to perform only one forward pass on the base model, whereas  GRPO needs to perform one forward pass on the base model, one forward pass on the reference model, and one backward pass (backpropagation) on the policy. Following the standard estimation in literature, for a model with P parameters and a sequence length of L, a single forward pass requires approximately 2PL FLOPs, and a backward pass requires 4PL FLOPs. Consequently, ES needs 2PL FLOPs for each sample evaluation, whereas GRPO needs 2+2+4=8PL FLOPs. Note that for  earlier RL methods such as PPO, the needed FLOPs are even larger since there are additional forward and backward passes for the critic model.
>
> To further evaluate the run time difference empirically, we run ES and GRPO using exactly the same hardware (6 H200 GPUs in the same machine). For a total number of $3\times 10^6$ sample evaluations on the countdown task (same as in the paper), the wall-clock time of ES and GRPO were:
> ||ES|GRPO|
> |-|-|-|
> |Qwen2.5-1.5B|2h 48m|16h 58m|
> |Qwen2.5-3B|5h 35m|25h 12m|
>
> Thus, the wall-clock time is significantly lower for ES compared to GRPO.
>
> 4. Q1. In principle, ES can indeed be combined with RL since they search in different spaces (parameter space vs. action space) and use different learning mechanisms (gradient-free vs. gradient-based). For global exploration, perturbation-based parameter space search should perform better since it is more explorative and less likely to get stuck in an initial local optimum. For fine-grained local search, gradient-based methods should be more effective, though their computational cost is higher than that of gradient-free methods. Using ES for exploration and RL for exploitation is a viable pipeline and a promising direction for future research. We will add this direction in Future Work; thanks for the suggestion.
>
> 5. Q2. The most obvious way is to use ES to replace RL in the post-pretraining pipeline,  for instance to enhance reasoning capabilities or improve alignment. ES only needs to perform inferences to obtain response-level scalar rewards, so the hardware requirement is even lower than that of RL. However, as mentioned in response to Q1, it is also possible to apply ES to strengthen current RL fine-tuning, e.g. run ES first for exploration then RL for exploitation, which is an interesting direction for future work.
>
> 6. Q3. To investigate this question, we ran a new experiment to further scale ES to larger models (on the countdown task with population size 30):
>
> Results for Qwen2.5-32B-Instruct:
> ||Math500|Miverva|Olympiad|AIME|AMC|
> |-|-|-|-|-|-|
> |Base Model|0.61|0.35|0.29|0.13|0.33|
> |After ES|0.80|0.40|0.44|0.30|0.54|
>
> Results for Qwen2.5-72B-Instruct:
> ||Math500|Miverva|Olympiad|AIME|AMC|
> |-|-|-|-|-|-|
> |Base Model|0.66|0.36|0.39|0.23|0.38|
> |After ES|0.83|0.40|0.47|0.27|0.59|
>
> The main result is that ES consistently improves over 32B and 72B models. Based on the results so far, we do not see a bottleneck in scaling up model sizes with a small population size. This may be due to the lower intrinsic dimensionality of larger pre-trained models (see Response W5 to Reviewer QJMr). Since ES does not use gradients, its implementation to larger models should be easier than gradient-based methods like RL. Such scaling is an interesting direction of further work.

---

> > ### Author Rebuttal · Reviewer_8oKX · 2026-04-02
> >
> > Thank the authors for the clarification, I also read other reviews and the responses. Although there is still some minor weakness in the evaluation part, it does not affect the soundness and the originality of this paper in general. Therefore, I would keep my positive evaluation.

---

> > > ### Author Response · Authors · 2026-04-02
> > >
> > > Thank you for your time and effort in reviewing our work. Your constructive comments and recognition of our work are very much appreciated.

---

### Decision · Program_Chairs · 2026-04-30

**Decision:**

Accept (regular)

**Comment:**

The paper demonstrates the application of evolutionary strategies for full fine-tuning of billion-scale LLMs, achieving competitive performance or outperforming state-of-the-art RL baselines. Reviewers raised some concerns in their initial reviews, such as the computational overhead of the method, which were addressed in the rebuttal. While the evaluation is somewhat limited, I agree with the reviewers that this is an interesting and promising paper and recommend acceptance.